# Stabilizing hidden room-temperature ferroelectricity via a metastable atomic distortion pattern

Jeong Rae Kim[1,2,9], Jinhyuk Jang[3,9], Kyoung-June Go[3], Se Young Park [1,2,4✉], Chang Jae Roh[5], John Bonini[6], Jinkwon Kim[1,2], Han Gyeol Lee[1,2], Karin M. Rabe[6], Jong Seok Lee[5], Si-Young Choi [3✉], Tae Won Noh [1,2✉] & Daesu Lee [7,8✉]

Nonequilibrium atomic structures can host exotic and technologically relevant properties in otherwise conventional materials. Oxygen octahedral rotation forms a fundamental atomic distortion in perovskite oxides, but only a few patterns are predominantly present at equilibrium. This has restricted the range of possible properties and functions of perovskite oxides, necessitating the utilization of nonequilibrium patterns of octahedral rotation. Here, we report that a designed metastable pattern of octahedral rotation leads to robust room-temperature ferroelectricity in $CaTiO_3$, which is otherwise nonpolar down to 0 K. Guided by density-functional theory, we selectively stabilize the metastable pattern, distinct from the equilibrium pattern and cooperative with ferroelectricity, in heteroepitaxial films of $CaTiO_3$. Atomic-scale imaging combined with deep neural network analysis confirms a close correlation between the metastable pattern and ferroelectricity. This work reveals a hidden but functional pattern of oxygen octahedral rotation and opens avenues for designing multifunctional materials.

[1] Center for Correlated Electron Systems, Institute for Basic Science (IBS), Seoul 08826, Korea. [2] Department of Physics and Astronomy, Seoul National University, Seoul 08826, Korea. [3] Department of Materials Science and Engineering, Pohang University of Science and Technology (POSTECH), Pohang 37673, Korea. [4] Department of Physics, Soongsil University, Seoul 07027, Korea. [5] Department of Physics and Photon Science, Gwangju Institute of Science and Technology (GIST), Gwangju 61005, Korea. [6] Department of Physics and Astronomy, Rutgers University, Piscataway, NJ 08854-8019, USA. [7] Department of Physics, Pohang University of Science and Technology (POSTECH), Pohang 37673, Korea. [8] Asia Pacific Center for Theoretical Physics, Pohang 37673, Korea. [9]These authors contributed equally: Jeong Rae Kim, Jinhyuk Jang. ✉email: sp2829@ssu.ac.kr; youngchoi@postech.ac.kr; twnoh@snu.ac.kr; dlee1@postech.ac.kr

Despite their simple structure, $ABO_3$ perovskites offer a wide range of functionalities including superconductivity, metal–insulator transition, ferroelectricity, and ferromagnetism. These functionalities largely originate from the rotation of corner-connected $BO_6$ octahedra (Fig. 1a), which allows the $ABO_3$ family to host a variety of cations. Furthermore, the angle and pattern of oxygen octahedral rotation (OOR)[1] directly influence electrical conductivity[2], magnetic superexchange interaction[3], dielectric properties[4], and so on. While this allows OOR to serve as a fundamental parameter for understanding functional perovskites, most perovskite bulks adopt the orthorhombic ($Pnma$) structure, corresponding to the $a^-b^+a^-$ OOR pattern in Glazer notation (Fig. 1a, b)[5]. Such predominance of the $a^-b^+a^-$ OOR pattern has prevented full exploitation of functional perovskites.

In particular, the most common $a^-b^+a^-$ OOR pattern has been shown to compete with an important functional property, namely ferroelectricity[4]. As a result, most $Pnma$ perovskites remain paraelectric, even down to 0 K, whereas their cubic phases could have instability for both ferroelectric and OOR distortions. This has motivated recent attempts to utilize nonequilibrium OOR, e.g., via artificial heteroepitaxy[6,7]. Despite extensive works[8–11], however, there is still a lack of studies on engineering the pattern itself of OOR and then generating functionalities.

In this work, we designed a means to stabilize a nonequilibrium OOR pattern of $CaTiO_3$ in oxide heterostructures. Unlike the original $a^-b^+a^-$ pattern, the metastable OOR pattern appears to be compatible with ferroelectricity, leading to the emergence of room-temperature ferroelectricity.

Moreover, we combined transmission electron microscopy techniques with deep neural network analysis and revealed the strong coupling between the metastable OOR pattern and ferroelectricity. Our work suggests that engineering the nonequilibrium OOR pattern unveils the hidden functionalities of materials.

## Results

**Theoretical search for nonequilibrium OOR patterns of $CaTiO_3$.** Taking $CaTiO_3$—the first discovered perovskite compound—as a model system, we explore the feasibility of achieving ferroelectricity via its nonequilibrium OOR patterns (Supplementary Fig. S1). Bulk $CaTiO_3$ has the $Pnma$ space group symmetry and the $a^-b^+a^-$ OOR pattern below 1512 K (ref. [12]) and exhibits a stable nonpolar, paraelectric phase down to 0 K; bulk $CaTiO_3$ exhibits antipolar Ca displacements. It is notable that $CaTiO_3$ is an incipient ferroelectric material with a negative Curie–Weiss temperature[13–15], so that it has potential to be engineered into a ferroelectric phase. Of the 23 possible OOR patterns, we focus on 10 patterns, since the other patterns rarely occur in a single $ABO_3$ compound[5]. According to our density-functional theory (DFT) calculations (Fig. 1c), the nonpolar $Pnma$ with the $a^-b^+a^-$ OOR pattern has the lowest energy, consistent with the bulk crystal structure of $CaTiO_3$. The highest energy is seen in the cubic $Pm\bar{3}m$ without OOR (i.e., $a^0a^0a^0$), which is highly unstable due to the small size of the A-site Ca ion[16]. The other eight OOR patterns are located in between and, to the best of our knowledge, have not been reported in bulk

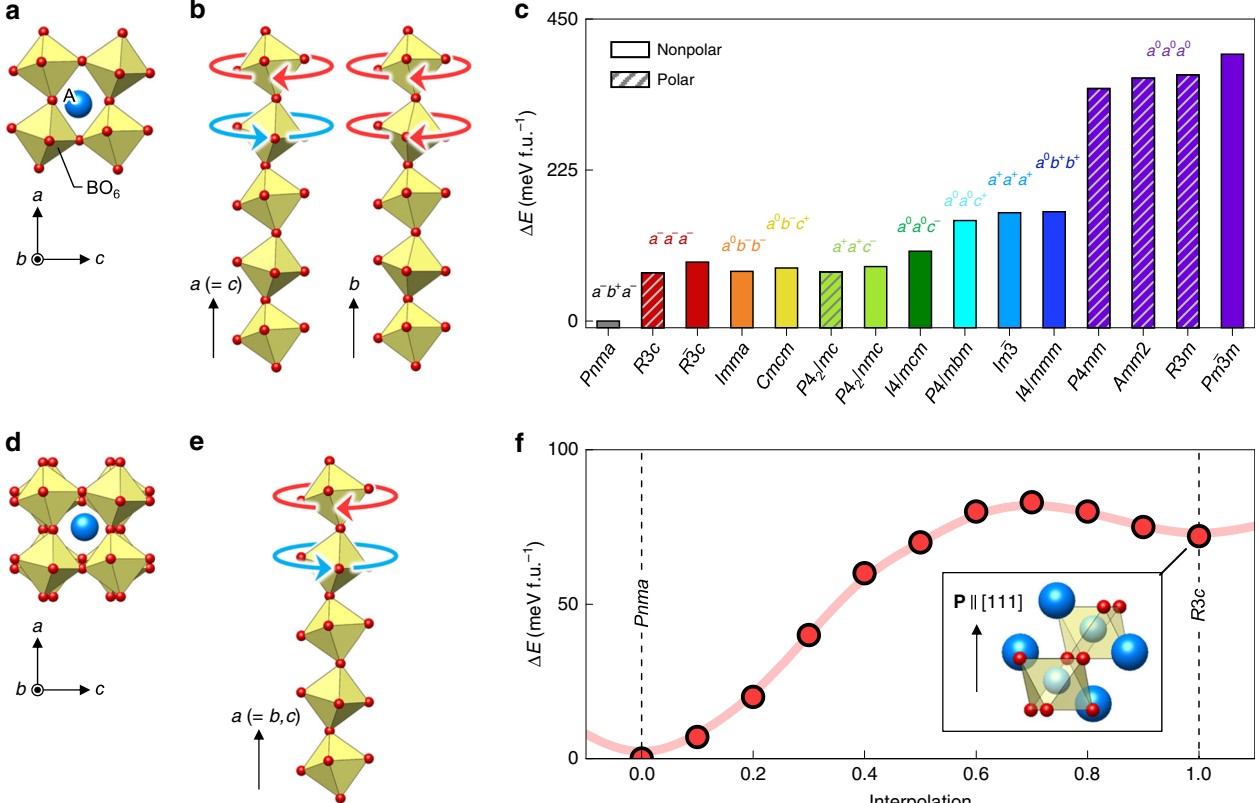

**Fig. 1 Hidden ferroelectricity of $CaTiO_3$ in its metastable oxygen octahedral rotation (OOR) pattern. a** Atomic structure of $ABO_3$ perovskite with an $a^-b^+a^-$ OOR pattern ($Pnma$). **b** $a^-b^+a^-$ OOR pattern, showing out-of-phase rotation along the $a$- and $c$-axes and in-phase rotation along the $b$-axis. **c** Density-functional theory (DFT) calculation of the energy landscape for various OOR patterns of $CaTiO_3$. The total energy for $Pnma$ is set to zero. Five structures ($R3c$, $P4_2/mc$, $P4mm$, $Amm2$, and $R3m$) are predicted to be polar. **d** Atomic structure of $ABO_3$ perovskite with an $a^-a^-a^-$ OOR pattern ($R3c$ or $R\bar{3}c$). **e** $a^-a^-a^-$ OOR pattern, showing out-of-phase rotation along all the axes. **f** Energy barrier between the $Pnma$ and metastable $R3c$ phases of $CaTiO_3$. We interpolate the transition between the $Pnma$ and $R3c$ structures by smoothly changing the OOR along the $b$-axis.

CaTiO$_3$. Our theory predicts that among the eight OOR patterns, $a^-a^-a^-$ and $a^+a^+c^-$ allow for polar structures with the space groups $R3c$ and $P4_2/mc$, respectively. Given their moderate energy cost (as small as 100 meV f.u.$^{-1}$), our DFT calculations indicate the possibility to thermodynamically stabilize the nonequilibrium $a^-a^-a^-$ or $a^+a^+c^-$ pattern and then achieve ferroelectricity in CaTiO$_3$.

Here, we take $a^-a^-a^-$ (Fig. 1d, e) as the target OOR pattern of CaTiO$_3$ due to its experimental accessibility, which is discussed later. Our first-principles calculation finds no unstable phonon mode in the polar $R3c$ with the $a^-a^-a^-$ OOR pattern (Supplementary Fig. S3), indicating its metastability (Fig. 1f). The calculated polarization value, $P$, of the $R3c$-CaTiO$_3$ is 44 μC cm$^{-2}$, and the energy barrier, $\Delta E$, for polarization switching is 16 meV f.u.$^{-1}$. These values are comparable to those of archetypal ferroelectric perovskites[17], such as BaTiO$_3$ ($P = 20$ μC cm$^{-2}$, $\Delta E = 11.6$ meV f.u.$^{-1}$) and PbTiO$_3$ ($P = 78$ μC cm$^{-2}$, $\Delta E = 32.6$ meV f.u.$^{-1}$), whose ferroelectric transition temperatures are 393 and 760 K, respectively. Therefore, artificial stabilization of the metastable $a^-a^-a^-$ OOR pattern may result in robust room-temperature ferroelectricity—hitherto hidden at equilibrium—in CaTiO$_3$.

**Heteroepitaxial stabilization of $R3c$-CaTiO$_3$.** While the $R3c$-CaTiO$_3$ is metastable in bulk, with a calculated energy barrier of 10 meV f.u.$^{-1}$ (Fig. 1f), heteroepitaxial growth could make it more stable. Through simply modulating OOR angles[18], it might be difficult to covert the OOR pattern itself from $a^-b^+a^-$ (i.e., corresponding to $Pnma$) to $a^-a^-a^-$ (i.e., corresponding to $R3c$). To overcome this, we constrain CaTiO$_3$ to the pseudo-cubic (111)$_{pc}$ plane of the LaAlO$_3$ substrate, which has the $a^-a^-a^-$ OOR pattern (note that we use a pseudo-cubic unit cell throughout and omit the pc subscript hereafter). Figure 2a, b emphasize that on the (111) plane, $a^-b^+a^-$ and $a^-a^-a^-$ OORs lead to disparate lattice symmetries; among the ten possible OOR patterns, only the $a^-a^-a^-$ pattern allows for a regular hexagon network of A-site ions. Furthermore, the (111) interface maximizes the octahedral connectivity through three metal-oxygen-metal

bonding (Supplementary Fig. S5), so that the OOR pattern of CaTiO$_3$ could more likely follow that of LaAlO$_3$. Accordingly, coherent, epitaxial growth of CaTiO$_3$ films on an LaAlO$_3$ (111) substrate could effectively stabilize the $a^-a^-a^-$ OOR pattern (Fig. 2c) in CaTiO$_3$.

We fabricate high-quality CaTiO$_3$ films on an LaAlO$_3$ (111) substrate using a pulsed laser deposition (PLD) technique (see "Methods"). As a nonequilibrium growth method[19], PLD could assist in achieving the metastable $a^-a^-a^-$ OOR pattern in CaTiO$_3$ films. Atomic force microscopy (Supplementary Fig. S8b) and X-ray diffraction (XRD) (Supplementary Fig. S8c) confirm that the films have smooth surfaces and high crystallinity. Previous works have reported experimental difficulties in achieving sharp, (111)-oriented heterointerfaces of perovskite oxides due to their high surface polarity[20]. In this work, we utilize in situ annealing of an LaAlO$_3$ substrate in ultra-high vacuum to achieve an atomically sharp CaTiO$_3$/LaAlO$_3$ (111) interface ("Methods" and Supplementary Fig. S9)[21]. Given this sharp interface, the OOR pattern in CaTiO$_3$ (111) films could follow that (i.e., the $a^-a^-a^-$ pattern) of the underlying LaAlO$_3$ substrate.

**Optical characterization of polar symmetry in CaTiO$_3$ (111) films.** Using optical second harmonic generation (SHG) polarimetry, we identify the macroscopic polar point-group symmetry of CaTiO$_3$ (111) films. Figure 2d shows a clear SHG signal in CaTiO$_3$ (111) films, which best fits with the polar point group of $3m$. This is consistent with our prediction that the CaTiO$_3$/LaAlO$_3$ (111) heteroepitaxy could stabilize the polar $R3c$ phase with an $a^-a^-a^-$ OOR pattern in CaTiO$_3$, belonging to the polar point group of $3m$. Furthermore, the SHG studies show a typical thickness dependence of ultrathin ferroelectric films. While we observe a persistent SHG signal in films thicker than 2.6 nm (i.e., 12 unit cells), the SHG signal disappears in films thinner than 1.3 nm (6 unit cells; Supplementary Fig. S11). This size effect is an intrinsic characteristic in ultrathin ferroelectrics[22], wherein ferroelectricity disappears at a finite critical thickness, e.g., due to depolarization field. Therefore, the observed SHG signal should

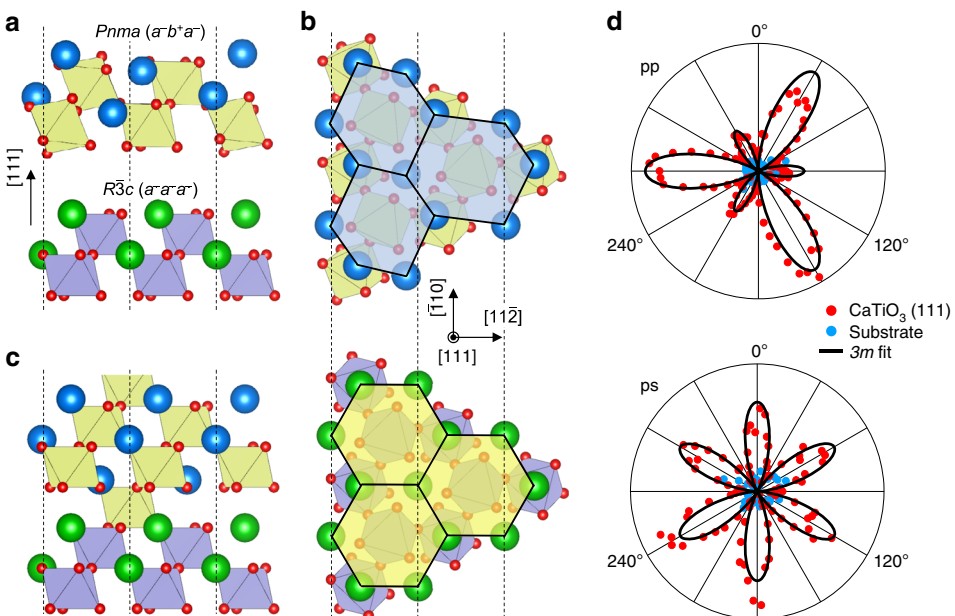

**Fig. 2 Heteroepitaxial stabilization of the $a^-a^-a^-$ OOR pattern in CaTiO$_3$. a** Structural mismatch between the $a^-b^+a^-$ and $a^-a^-a^-$ OOR patterns in the (111) plane. **b** Disparate in-plane lattice symmetries of the $a^-b^+a^-$ and $a^-a^-a^-$ OOR patterns in the (111) plane. **c** Schematic diagram of OOR pattern engineering in the (111)-oriented heterointerface between $a^-b^+a^-$ and $a^-a^-a^-$ structures. **d** Optical second harmonic generation (SHG) signals from 2.6 nm-thick CaTiO$_3$ (111) film (red) and LaAlO$_3$ substrate (blue). "p" ("s") indicates parallel (perpendicular) polarization of light with respect to the plane of incidence.

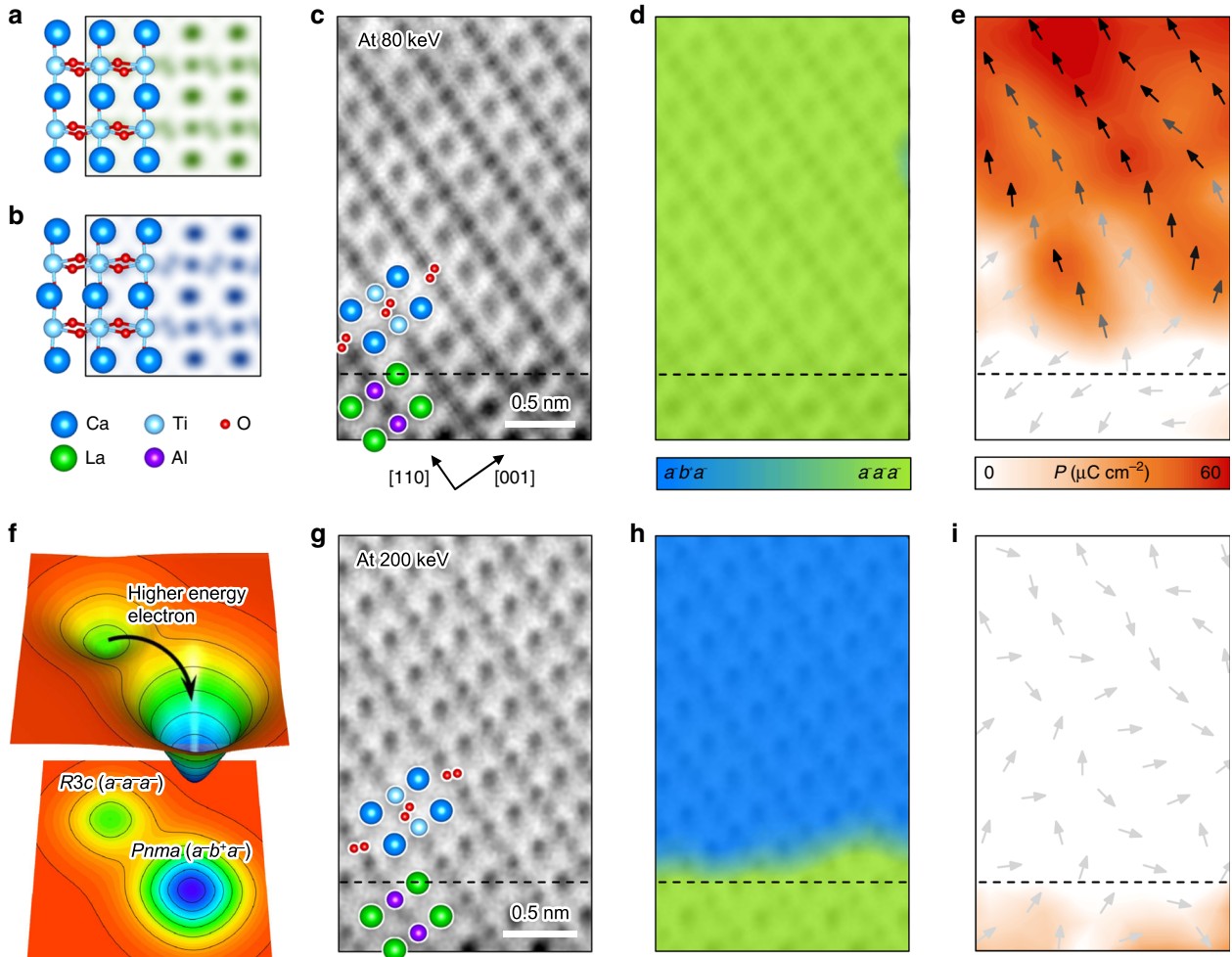

**Fig. 3 Atomic-scale analysis of the OOR pattern and polarity in CaTiO₃ films.** Simulated annular bright-field scanning transmission electron microscopy (ABF-STEM) images along the zone axis of [1$\bar{1}$0] for $a^-a^-a^-$ (**a**) and [101] for $a^-b^+a^-$ (**b**) OOR patterns. **c–i** (**c**) and (**g**) show ABF-STEM images of 2.6 nm-thick CaTiO₃ (111) film along the zone axis of [1$\bar{1}$0], measured using 80-keV (**c**) and 200-keV (**g**) electron kinetic energy. **d**, **h** display the maps of OOR patterns, identified by deep neural network analysis, in the same regions as in (**c**) and (**g**), respectively. Color indicates the probability of each OOR pattern. **e**, **i** present polarization vectors for each unit cell of the same regions as in (**c**) and (**g**), respectively. Arrows denote the polarization direction; the stronger the polarization, the darker the arrow color. Strength of polarization is also expressed as a color map, ranging from white (weak) to red (strong). **f** illustrates a schematic free energy landscape, showing the relaxation of $R3c$ state ($a^-a^-a^-$) into $Pnma$ state ($a^-b^+a^-$).

arise solely from the bulk of CaTiO₃ (111) films, rather than from their surfaces and interfaces.

**Atomic-scale imaging of OOR patterns and polarization.** Scanning transmission electron microscopy (STEM) can provide atomic-scale information on the OOR pattern, as well as ferroelectricity, of the films. In particular, annular bright-field STEM (ABF-STEM) images are sensitive to light atoms (e.g., oxygen)[23], making them an effective tool for visualizing OOR patterns. Because OOR generally causes elongation of oxygen peaks in ABF-STEM images, qualitative analysis of such elongated oxygen peaks can provide the means to identify the OOR pattern in the films[24]. Indeed, our ABF-STEM simulations, based on the calculated atomic structures in Fig. 1c, indicate that each OOR pattern has its unique oxygen peak feature in the [1$\bar{1}$0] projection (Supplementary Fig. S12): the $a^-a^-a^-$ OOR pattern has the elongated oxygen peaks aligned in one crystalline direction (Fig. 3a) and the $a^-b^+a^-$ pattern has the elongated oxygen peaks arranged in a zigzag-like pattern (Fig. 3b). Importantly, this

unique feature of the oxygen peak shapes allows the OOR pattern of the CaTiO₃ films to be identified.

Furthermore, we employ a deep neural network approach to identify the OOR patterns from the measured ABF-STEM images (see "Methods")[25,26]. Identification of OOR patterns could be ambiguous, e.g., due to nontrivial experimental noise, difficulty in the quantification of shape information, or infinitesimal differences among OOR shapes. The deep neural network approach, however, can help us to identify OOR patterns beyond human cognition. Figure 3c displays the measured ABF-STEM image of CaTiO₃ (111) films with a zone axis of [1$\bar{1}$0], exhibiting oxygen peaks elongated along the same direction. This is consistent with the simulated ABF-STEM image for the $a^-a^-a^-$ OOR pattern (Fig. 3a). Our deep neural network analysis indeed identifies the $a^-a^-a^-$ OOR pattern from the measured ABF-STEM image of CaTiO₃ (111) films (Fig. 3d). The ABF-STEM image also confirms that such an $a^-a^-a^-$ OOR pattern coexists with electric polarization in CaTiO₃ (111) films (Fig. 3e), which is in good agreement with our theoretical prediction and SHG results. [Fig. 3e shows rather suppressed electric polarization near the

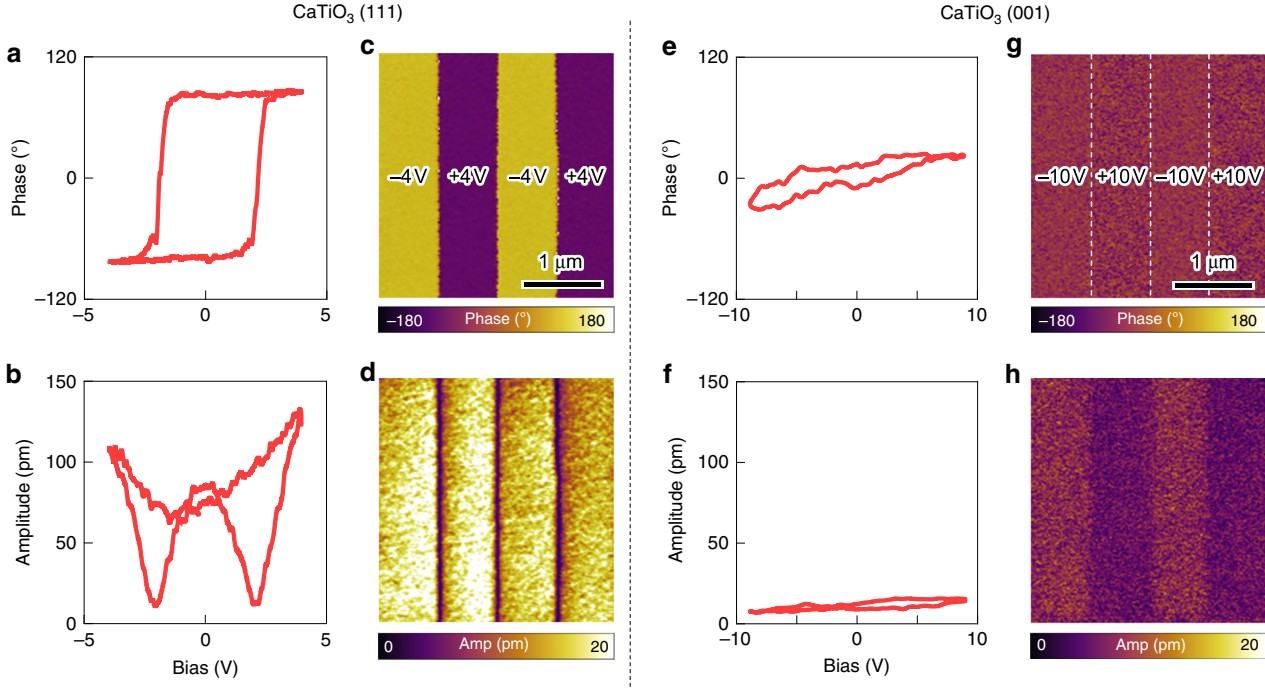

**Fig. 4 Switchable and stable polarization in CaTiO₃ (111) films.** Switching behavior of the piezoelectric phase (**a**) and amplitude (**b**) in 2.6 nm-thick CaTiO₃ (111) film. Bipolar domain patterns, written on the same CaTiO₃ (111) film. Piezoelectric phase (**c**) and amplitude (**d**) show two stable ferroelectric domains. Nonferroelectric behavior of the piezoelectric phase (**e**) and amplitude (**f**) in 3.8 nm-thick CaTiO₃ (001) film. The absence of domain patterns in the same CaTiO₃ (001) film. Piezoelectric phase (**g**) and amplitude (**h**) show the absence of ferroelectric domains.

bottom interface, which might originate from interface dipoles (Supplementary Fig. S6).] In stark contrast, CaTiO₃ (001) films, which differ from CaTiO₃ (111) films only in their crystalline orientation, show the $a^-b^+a^-$ OOR pattern and no electric polarization (Supplementary Fig. S13i–l). This is consistent with the nonpolar *Pnma* phase of bulk CaTiO₃ (Fig. 3b). These results thus firmly validate our heteroepitaxial design principle for stabilizing the nonequilibrium OOR pattern and achieving polar CaTiO₃ at room temperature.

**Demonstration of switchable polarization.** Then, using piezoresponse force microscopy (PFM), we examine whether the electric polarization is switchable and stable—the signature of ferroelectricity—at room temperature. To apply an electric field, we prepare CaTiO₃ films on an LaAlO₃ substrate buffered with LaNiO₃ films as the bottom electrode. Given that LaNiO₃ has the $R\bar{3}c$ structure with an $a^-a^-a^-$ OOR pattern, the same heteroepitaxy control of the OOR patterns in CaTiO₃ (Fig. 2c) would work. Our PFM measurement reveals clear hysteretic behavior (Fig. 4a, b), with a phase difference of around 180° in CaTiO₃ (111) films, consistent with the typical ferroelectric responses under PFM. In contrast, CaTiO₃ (001) films show almost no change in the PFM phase, and a negligible PFM amplitude (Fig. 4e, f); accordingly, we exclude extrinsic electrochemical effects[27] as the origin of the PFM response in (111) films. Figure 4 verifies that the [111]-directed film growth is essential in stabilizing the ferroelectric state, as it not only guarantees the unique geometric constraint (Fig. 2), but also maximizes octahedral connectivity (Supplementary Fig. S5). Bipolar domain patterns (Fig. 4c, d) were writable and stable on CaTiO₃ (111) films. The pristine domain structure of CaTiO₃ (111) films was difficult to define (Supplementary Fig. S14); however, it may be attributable to the ferroelectric domain structure (Supplementary Fig. S18),

which can assume a mono-domain configuration upon electrical poling (Fig. 4c, d).

We examined the stability of the written domains by monitoring the PFM signal as a function of time (Supplementary Fig. S15). The PFM signal gradually decays, following a power-law with a decay exponent of 0.145. Although comparable decay exponents have been reported in the relaxor ferroelectrics[28], a similar relaxation has also been observed in prototypical ferroelectric films at the ultrathin limit[29]. Based on our comprehensive DFT and STEM studies, we attribute the relaxation behavior of the ferroelectric $R3c$-CaTiO₃ to the huge depolarization field in ultrathin ferroelectric films. Taken together, our theoretical and experimental results consistently demonstrate that artificial stabilization of the nonequilibrium OOR pattern leads to robust room-temperature ferroelectricity in otherwise nonpolar CaTiO₃ films.

**Discussion**
Our results highlight that we achieve both selective stabilization and atomic-scale imaging of the ferroelectrically active, nonequilibrium $a^-a^-a^-$ OOR pattern. Importantly, because nonequilibrium states inherently involve complex structural instabilities, it has been difficult to selectively stabilize them and resolve their atomic structures[30]. However, by using the moderate conditions in the STEM experiments (e.g., electron kinetic energy of 80 keV), we successfully resolve the artificially stabilized $a^-a^-a^-$ OOR pattern at the atomic scale, without structural relaxation to the ground state ($a^-b^+a^-$ pattern). This implies that STEM imaging with higher electron energy would cause greater perturbation of the system, possibly resulting in structural relaxation from the $a^-a^-a^-$ to the $a^-b^+a^-$ OOR pattern (Fig. 3f). Indeed, when imaged with a higher electron kinetic energy (200 keV), the atomic structure relaxes to that of the $a^-b^+a^-$ OOR pattern, identified via deep neural network analysis (Fig. 3g, h).

Importantly, this relaxation of the OOR pattern coincides with the disappearance of electric polarization (Fig. 3i). Furthermore, even after the OOR pattern is relaxed to $a^-b^+a^-$, CaTiO$_3$ still remains fully strained to the underlying LaAlO$_3$ substrate (Supplementary Fig. S16). This emphasizes that the epitaxial strain itself cannot stabilize the $a^-a^-a^-$ OOR pattern and resulting ferroelectricity in CaTiO$_3$/LaAlO$_3$ heterostructures, consistent with our DFT calculations (Supplementary Fig. S4). Thus, these results not only confirm a close correlation between the metastable $a^-a^-a^-$ OOR pattern and ferroelectricity, but also exclude the epitaxial strain as a primary origin for the ferroelectricity in CaTiO$_3$ (111) films.

Interest in understanding and utilizing metastable or hidden states, which would allow for exotic phenomena and functions, has been growing[30–34]. While optical pumping has been mainly used to explore nonequilibrium transient states[33], we stabilize a nonequilibrium OOR pattern and then achieve robust ferroelectricity. Our approach to utilize metastable OOR patterns for functional perovskite oxides is distinct from conventional chemical substitution[35] or strain[36]/dimensional[29] engineering. Moreover, our theory suggests that the metastable OOR pattern may facilitate the development of multifunctionalities (Supplementary Fig. S17): our results indicate that in the metastable $a^-a^-a^-$ OOR pattern, an increase in rotation angle could enhance the electric polarization, in stark contrast to the conventional $a^-b^+a^-$ pattern (which is competitive with ferroelectricity). Such cooperation between the OOR and polar distortion could constitute a generic principle for polarizing various material systems, including not only dielectrics but also conductors. Thus, OOR pattern engineering is expected to be widely used to design multifunctional perovskite oxides, in which the broken inversion symmetry is combined with electron conduction, magnetism, and topological phases.

## Methods

**Thin film growth.** Commercially available LaAlO$_3$ single crystal substrates (Crystec GmbH, Germany) were used. Prior to the growth, LaAlO$_3$ (111) and (001) substrates were dipped in deionized water and sonicated for 2 h. Then, the substrates were annealed in the growth chamber in situ. The annealing temperature, background gas pressure, and annealing time were 1000 °C, <1.0 × 10$^{-7}$ mTorr, and 1 h, respectively. Using this procedure for preparing substrates, we realized clean (111)-oriented heterointerfaces despite the high surface polarity (Fig. S9). We observed no structural defects or interfacial disorder over a large area (Fig. S9a, d)[20]. The small-scale images (Fig. S9b, c, e, f) confirm that the interfacial atomic structure is abrupt and epitaxial.

CaTiO$_3$ and LaNiO$_3$ thin films were grown using PLD. For the growth of thin films, the substrate temperature and background oxygen partial pressure were kept at 600 °C and 10 mTorr, respectively. Polycrystalline Ca$_{1.1}$TiO$_{3.1}$ and LaNiO$_3$ targets were ablated with a KrF excimer laser (LPXpro, Coherent, USA). Off-stoichiometry of the cation in oxide dielectrics can produce defect dipoles and ferroelectricity[29], which are undesirable in this work. To exclude such extrinsic properties, we optimized the growth of cation-stoichiometric CaTiO$_3$ thin films. In the growth of CaTiO$_3$ thin films, the laser energy density played a critical role in determining film quality. Figure S7a–f shows the surface morphology and reflection high energy electron diffraction (RHEED) patterns of CaTiO$_3$/LaAlO$_3$ (001) thin films grown with the laser energy density of 0.5, 1.0, and 1.5 J cm$^{-2}$. We noticed that a smooth surface and spotty RHEED pattern are only achieved with an optimal laser energy density of 1.0 J cm$^{-2}$. When the value is too low, cracked surfaces can occur. On the other hand, when it is too high, rough surfaces and extra peaks in RHEED patterns have been reported. This feature is consistent with the off-stoichiometric CaTiO$_3$ thin film, reported previously[37,38]. In addition, the CaTiO$_3$ thin film grown on a (LaAlO$_3$)$_{0.3}$–(Sr$_2$AlTaO$_6$)$_{0.7}$ (LSAT) (001) substrate with optimal laser energy density has the same $c$-axis lattice parameter as films grown by molecular beam epitaxy (Fig. S7g). We examined the possible oxygen vacancy formation in our optimal CaTiO$_3$ film by postannealing in oxygen. The postanneal temperature, background gas pressure, and annealing time were 500 °C, 760 Torr of pure oxygen, and 2 h, respectively. This postannealing process did not change the lattice parameter of our optimal CaTiO$_3$, implying that the sample is already well-oxidized. Thus, our CaTiO$_3$ thin films grown with optimal laser energy density should be reasonably stoichiometric in terms of cation and oxygen. Throughout this work, we used the optimal laser energy density for growing CaTiO$_3$ thin films.

We fabricated CaTiO$_3$/LaAlO$_3$ (111) heterostructures using the same growth condition as for CaTiO$_3$/LaAlO$_3$ (001). The in situ RHEED intensity monitoring shown in Fig. S8a indicates layer-by-layer growth of CaTiO$_3$ on the LaAlO$_3$ (111) substrate. The grown films had smooth surfaces with step and terrace structures (Fig. S8b).

**Characterization of thin films.** XRD (AXS D8 with a Vantec line-detector, Bruker, USA) was used for structural analysis of thin films. All films were epitaxially grown on the LaAlO$_3$ (111) substrates and clear Kiessig fringes are seen (Fig. S8c). For ultrathin films with a thickness of several nanometers, we used Huber six-circle diffractometers at Sector 3A of the Pohang Light Source.

Switchable ferroelectric polarization was confirmed using scanning probe microscopy (Cypher, Asylum Research, UK) with conductive probes (PPP-EFM, Nanosensors). Dual AC resonance tracking PFM was used to read/write bipolar domain patterns and obtain $d_{33}$ values as a function of the applied voltage. To exclude the contribution of electrostatic force to PFM signals, PFM was performed at higher-harmonic resonance frequency near 900 kHz.

Switching spectroscopy PFM measurements were also carried out on CaTiO$_3$/LaNiO$_3$/LaAlO$_3$ (001) films as well. Bias-dependent PFM phase and amplitude measurements show no hysteresis loop up to the maximum bias voltage of 10 V (Fig. 4e, f). We attempted to write bipolar domains as in the case of (111)-oriented CaTiO$_3$ (Fig. 4g, h). However, the phase difference between the two domains was small and the amplitude of the two domains was not equivalent; this is characteristic of electrostatic charge injection rather than ferroelectricity, which further corroborates the absence of ferroelectricity in (001)-oriented CaTiO$_3$ films.

**Density-functional theory calculation.** We performed first-principles DFT calculations within the local density approximation[39,40] using the Vienna ab initio simulation package (VASP)[41,42]. The projector augmented wave method[43] was used with an energy cut-off of 500 eV. The Brillouin zone was sampled with an 8 × 8 × 8 $k$-point grid for the 5-atom unit cell of rhombohedral CaTiO$_3$ and a 4 × 4 × 4 $k$-point grid is used for a 2 × 2 × 2 supercell with 40 atoms to accommodate the relevant octahedral rotations and polar distortions. Convergence was reached if the consecutive energy difference is less than 10$^{-6}$ eV. The structural relaxation was conducted with a force threshold of 0.001 eV Å$^{-1}$. The polarization was calculated using the Berry-phase method[44] as implemented in VASP. The phonon dispersion was calculated using the density-functional perturbation theory implemented in the VASP and phonopy[45] with an increased energy cut-off (600 eV) and convergence threshold (10$^{-8}$ eV).

Locally stable polar structures for each of ten rotation patterns are obtained by evaluating the phonon dispersion of a 2 × 2 × 2 supercell (Fig. 1c). The ten rotation patterns were chosen based on reported rotation patterns of single-phase perovskite oxides[5]. The systematic search procedure to obtain locally stable polar structures is presented in Fig. S1.

With the $a^-a^-a^-$ OOR pattern, CaTiO$_3$ can either be polar $R3c$ or nonpolar $R\bar{3}c$ (Fig. S2). Based on the DFT calculation of LaAlO$_3$, Pnma-CaTiO$_3$, $R\bar{3}c$-CaTiO$_3$, and $R3c$-CaTiO$_3$, we estimated the lattice mismatch between CaTiO$_3$ and LaAlO$_3$. The results are summarized in Table S1. Overall, CaTiO$_3$ films are in small compressive strain on LaAlO$_3$ substrate.

We calculated phonon dispersion of CaTiO$_3$ with $R\bar{3}c$ and $R3c$ structures to check the ferroelectricity of CaTiO$_3$ in the given OOR pattern of $a^-a^-a^-$ (Fig. S3). $R\bar{3}c$-CaTiO$_3$ has imaginary polar modes at the $\Gamma$ point (Fig. S3a), so it must relax to other stable or locally stable structures. After following the flowchart in Fig. S1, $R\bar{3}c$-CaTiO$_3$ was relaxed to the $R3c$-CaTiO$_3$ rather than the ground state (Pnma-CaTiO$_3$). The $R3c$-CaTiO$_3$ did not show any unstable phonon modes (Fig. S3b). This local stability of $R3c$-CaTiO$_3$ implies that stabilization of the $a^-a^-a^-$ OOR pattern can realize the ferroelectricity in CaTiO$_3$.

To understand the role of epitaxial strain in the CaTiO$_3$/LaAlO$_3$ (111) heterostructures, we calculated the energy of Pnma and $R3c$ structures under biaxial strain (Fig. S4a). Under compressive or tensile strain up to 2%, Pnma always has lower energy than $R3c$. Furthermore, strain-dependent phonon dispersion calculations on the Pnma structure indicated the robust stability of the Pnma structure under biaxial strain (Fig. S4b–d). This excludes a significant role of the epitaxial strain in stabilizing $R3c$-CaTiO$_3$ in CaTiO$_3$/LaAlO$_3$ (111) heterostructures. Instead, the surface symmetry of LaAlO$_3$ (111) substrates should play the dominant role in stabilizing $R3c$-CaTiO$_3$.

We investigated the relation between OOR and ferroelectricity in $R3c$-CaTiO$_3$. Here, the OOR angle is defined by the amount of rotation perpendicular to the (111)-axis (Fig. S17a). The calculated strain- and OOR angle-dependent polarization is plotted in Fig. S17b. Overall, the polarization increases with the OOR angle, implying a cooperative coupling between OOR and ferroelectricity in $R3c$-CaTiO$_3$[46]. The cooperative coupling was enhanced under compressive strain, which was our experimental condition.

To explain the suppressed electric polarization of CaTiO$_3$ in the CaTiO$_3$/LaAlO$_3$ (111) interfacial region (Fig. 3e), we carried out DFT calculation of realistic heterostructure composed of $R3c$-CaTiO$_3$ and LaNiO$_3$. In the case of CaTiO$_3$/LaAlO$_3$ interface, we found that electrons or holes are doped in CaTiO$_3$ depending on interfacial termination configurations[47], which make our calculation complicated. Replacing LaAlO$_3$ with LaNiO$_3$ does not dope any charge carriers into the CaTiO$_3$ and will not affect our arguments on interface dipole. Figure S6 shows

the DFT calculation of $[(LaNiO_3)_{12}/(CaTiO_3)_{12}]$ superlattices with $Ni/CaO_3$ and $LaO_3/Ti$ interface terminations. Independent of interface termination, interface dipoles are always pointing from $CaTiO_3$ to $LaNiO_3$. When intrinsic polarization of $CaTiO_3$ is pointing the opposite direction (11th or 12th layer in Fig. S6c, d), polarization is suppressed in the interfacial region and it is consistent with our observation.

**Second harmonic generation experiment.** The symmetry of $CaTiO_3$ thin films grown on (001) and (111) $LaAlO_3$ substrate was characterized by SHG. Figure S10 displays the experimental set-up for the SHG. The 800-nm femtosecond wave, with an 80-MHz repetition rate and 30-fs duration, was irradiated on the thin films with an incidence angle of 45° as a fundamental wave (Vitara-T, Coherent). We focused the beam spot size into 1–30 μm. The polarization states of a fundamental and generated second harmonic wave were controlled to be p- or s-polarization by a half-wave plate and polarizer, respectively. To avoid the detection of the fundamental wave, we isolated the second harmonic wave using the short pass and bandpass filters. We monitored the intensity of the second harmonic response using a photomultiplier tube at the end of the optical path.

The allowed nonlinear susceptibility ($\chi$) components of point group $3m$ are $\chi_{xxx} = -\chi_{xyy} = -\chi_{yxy} = -\chi_{yyx}$, $\chi_{zxx} = \chi_{zyy}$, $\chi_{xxz} = \chi_{yyz}$, $\chi_{xzx} = \chi_{yzy}$, and $\chi_{zzz}$. We calculated and fitted the polarization-dependent SHG intensity from $CaTiO_3$ thin films, using the formulas below:

$$\begin{cases} I_{pp}(2\omega) = \left[(\chi_{zyy} - \chi_{yzy} - \chi_{yyz} + \chi_{zzz}) + \chi_{xxx}\sin(3\varphi)\right]^2 \\ I_{sp}(2\omega) = 4\left[\chi_{zyy} - \chi_{xxx}\sin(3\varphi)\right]^2 \\ I_{ps}(2\omega) = \left[\chi_{xxx}\cos(3\varphi)\right]^2 \\ I_{ss}(2\omega) = 4\left[\chi_{xxx}\cos(3\varphi)\right]^2 \end{cases} \quad . \quad (1)$$

We carried out SHG experiments on 2.6 and 1.3 nm-thick $CaTiO_3/LaAlO_3$ (111) thin films (Figs. 2d and S11). In the case of 2.6 nm-thick films, the observed SHG was well fitted with a point group $3m$. On the other hand, negligible SHG was detected in the 1.3 nm-thick films. Considering that most ferroelectrics lose their ferroelectricity below a finite thickness, it is expected that $CaTiO_3$ grown on $LaAlO_3$ (111) has a critical thickness for ferroelectricity between 1.3 and 2.6 nm. The critical thickness confirms that our observation is from neither the $CaTiO_3$ surface nor the $CaTiO_3/LaAlO_3$ interface. In addition, the possibility of defect-induced ferroelectricity, which is enhanced with reduced film thickness[29], is also ruled out.

**Scanning transmission electron microscopy.** The $CaTiO_3/LaAlO_3$ (111) sample has a metastable $R3c$ phase with an $a^-a^-a^-$ OOR pattern which can be relaxed to a $Pnma$ phase with an $a^-b^+a^-$ OOR pattern; thus, we used mild sample preparation and STEM imaging conditions for the $CaTiO_3/LaAlO_3$ (111) sample. Cross-sectional specimens oriented along the $[1\bar{1}0]$ direction of both samples were prepared by conventional mechanical flat polishing with less than 10 μm thickness. Mechanically polished samples were further milled using a 3 keV Ar ion beam for $CaTiO_3/LaAlO_3$ (001) and 2 keV for $CaTiO_3/LaAlO_3$ (111) and using a 0.1 keV Ar ion beam for removing surface damage from both samples (PIPS II, Gatan, USA). ABF-STEM images for light elements, i.e., oxygen, were acquired using STEM (ARM 200F, JEOL, Japan) with a spherical aberration corrector (ASCOR, CEOS GmbH, Germany) at Materials Imaging & Analysis Center of POSTECH in South Korea. Additionally, to minimize the electron beam-induced phase transition of the metastable $CaTiO_3/LaAlO_3$ (111) sample, imaging was also conducted at an 80-kV accelerating voltage with a convergence angle of 27 mrad.

The standard position of each unit cell was determined from the Ti- or Al-atomic column, and the positive charge position was defined by the six-neighboring Ca- or La-atomic column (because the Ca-atomic column off-centering is larger than that for Ti). The negative charge positions are disregarded under the assumption that the O and Ca ions shift in the opposite direction, and displacement vectors from all unit cells were calculated from the standard position to the positive charge center. Polarization, $P$, can be simply calculated by the following equation, $P = \frac{1}{V}\sum_i \delta_i Z_i$, where $V$ is the unit cell volume, and $\delta_i$ and $Z_i$ are the displacement length and effective charge of atom $i$, respectively. Born effective charges for $CaTiO_3$ and $LaAlO_3$ are adopted to reflect electronic and ionic polarization[48–50]. $V (= a^2c)$ was calculated from $a$ and $c$, which corresponds to the $\sqrt{2}[110]$ and $[001]$ Ca–Ca (or La–La) spacing of perovskite structure, respectively.

Figure S12 shows the simulated ABF-STEM images of the 10 OOR patterns of $CaTiO_3$ considered in Fig. 1c. All ABF-STEM simulations were conducted with the multi-slice method using Dr. Probe software (Ernst Ruska-Centre, Germany)[51], based on the atomic structures given by our DFT calculations. Figure S13 shows a comprehensive STEM data set of $CaTiO_3/LaAlO_3$ (111) and $CaTiO_3/LaAlO_3$ (001) heterostructures. Discussions on the $CaTiO_3/LaAlO_3$ (111) thin films are provided in the main text. For the $CaTiO_3/LaAlO_3$ (001) heterostructure, the bulk-like nonpolar $Pnma$ structure is clear. A high-quality hetero-interfacial structure is shown in Fig. S13i. Figure S13j shows the good agreement between the simulated ABF-STEM of $a^-b^+a^-$ and the structure of $CaTiO_3$. Note that both relaxed $CaTiO_3/LaAlO_3$ (111) (Fig. S13e) and $CaTiO_3/LaAlO_3$ (001) (Fig. S13i) have an

$a^-b^+a^-$ OOR pattern, but they are oriented along the $[101]$ and $[1\bar{1}0]$ directions, respectively. Convolutional neural network (CNN; Fig. S13k) and polarization mapping (Fig. S13l) evidence the $Pnma$ structure of $CaTiO_3$ grown on an $LaAlO_3$ (001) substrate.

$R3c$-$CaTiO_3$ films can exhibit domains, following the intrinsic twin domains of the underlying $LaAlO_3$ (111) single crystal substrate. We experimentally observed all four different domains of $R3c$-$CaTiO_3$, which is numbered in Fig. S18a. For the domains #1 and #4, we can easily observe the polarization in STEM because the polarization is perpendicular to the $[1\bar{1}0]$ zone axis (Fig. S18b). For the domains #2 and #3, however, polarization vectors have $[1\bar{1}0]$-components and we barely observe the polarization in STEM (Fig. S18b).

Given the threefold rotational symmetry of $LaAlO_3$ (111) substrate, there can be three equivalent domains of the relaxed $Pnma$-$CaTiO_3$ on $LaAlO_3$ (111). We refer to the three domains as $a^-b^+a^-$, $a^+b^-b^-$, and $a^-a^-c^+$ domains, respectively, as shown in Fig. S19. Since only $a^-b^+a^-$ shows the characteristic zigzag-like pattern of oxygen, we mainly discussed the $a^-b^+a^-$ in the main text.

**Deep neural network analysis.** The obtained STEM images were filtered to reduce background noise and to extract each atomic column position with sub-pixel accuracy. Denoising autoencoder (DAE), a deep machine learning technique for reconstructing images, was applied to avoid the image distortion caused by a conventional Fourier transformation filter. Each STEM image was sliced into image patches containing a single atom, as training or input data for DAE. Atomic image patches were reconstructed (output) through unsupervised training. The DAE was trained to minimize the difference between the input noisy image and the output reconstructed image. After a sufficient training process, DAE was trained to reconstruct only the atomic image, and not to imitate the noise from the input image. Reconstructed image patches with sharp atomic coordinate information were assembled to construct a full STEM image.

CNN, as a deep learning image recognition technique, was exploited for OOR pattern mapping. General CNNs are mainly composed of convolutional layer (Conv. Layer), max pooling, and fully connected layer (F.C. Layer) (Fig. S20). A Conv. Layer consists of several filters extracting the features from the input image. The filters are matrices and the elements are continuously renewed to learn which feature to extract during the training process. On the contrary, max pooling does not change during training; it compresses the feature images from previous steps. For efficiency, image compression reduces computing power consumption, and for performance, it desensitizes decision making to the small variance of image position. F.C. Layer is composed of several neurons, which make a definitive prediction of OOR, based on the extracted features. Neurons in F.C. Layer are matrices and change via training, like the filters in Conv. Layer.

The main challenge in the application of CNN to STEM imaging is the insufficient quantity of unambiguous training data. For this reason, simulated data were used for the training set. To exclude ambiguity, ABF-STEM simulation was conducted on DFT calculated structures along $[1\bar{1}0]$ projections of the $a^-a^-a^-$ and $a^-b^+a^-$ patterns in $CaTiO_3$. For the purpose of emulating real experimental conditions and improving the training data, simulated images were preprocessed[52] (Fig. S21). During the preprocessing steps, a $1 \times 2$ unit cell, which is the minimum criterion for OOR determination, was cropped to a random size in a random position, and then uniformly resized to $90 \times 72$ pixels, the input dimension for CNN. Resized images were convolved with probe sizes from 10 to 50 pm, which is the range between the ideal case (10 pm) and the technological limitation (50 pm). Lastly, artificial Gaussian noise was added with random values (between 40% and 60% of the data). These preprocessing steps reproduced 1350 images from a single simulation image.

For OOR identification through the information hidden by nontrivial noise, a custom deep neural network was devised (Fig. S22). This network possessed three Conv. Layers with 32, 64, and 64 filters ($5 \times 5$, $3 \times 3$ and $3 \times 3$, respectively) that slide across the whole of each input image. Max pooling layers were connected to the second and the third Conv. Layers. CNN ended up with two F.C. Layers, which had 128 and 2 neurons each. The latter layer implies a binary classification of $a^-a^-a^-$ or $a^-b^+a^-$ regardless of the projection of each OOR pattern.

For OOR mapping of the experimental ABF-STEM image, $1 \times 2$ unit cell images were cropped into image patches that slide across the whole image. Each image patch was resized to $90 \times 72$, which is the original input size of the custom CNN. Each patch returned its coordinates and OOR mapping result as a probability of each OOR pattern. Based on the coordinates, the patches were reintegrated with the color-mapped prediction result. The deep learning works and the calculations of atomic polarization were all performed based on Python. Also, Keras with TensorFlow backend was utilized as a deep learning library for training the model.

## Data availability

All relevant data presented in this paper are available from the authors upon reasonable request. Source data are provided with this paper.

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

## Acknowledgements

This work was supported by IBS-R009-D1 and by the National Research Foundation of Korea (NRF) grant funded by the Korea government (MSIT) (Nos. 2018R1A5A6075964 and 2019R1C1C1002558). S.-Y.C. mainly acknowledges the support of the Global Frontier Hybrid Interface Materials of the National Research Foundation of Korea (NRF) funded by the Ministry of Science and ICT (2013M3A6B1078872). D.L. and S.-Y.C. acknowledge the support by POSTECH-Samsung Electronics Industry-Academia Cooperative Research Center. S.Y.P. acknowledges the support by the National Research Foundation of Korea (NRF) grant funded by the Korea government (MSIT) (No. 2020R1F1A1076742). Work at GIST is supported by the Basic Science Research Program through the National Research Foundation of Korea (NRF) funded by the Ministry of Science, ICT and Future Planning (Nos. 2015R1A5A1009962, 2018R1A2B2005331). K. M.R. acknowledges the support from Office of Naval Research grant N00014-17-1-2770. Experiments at 3A beamline of PLS-II were supported in part by MSICT.

## Author contributions

D.L. conceived and designed the research. J.R.K. carried out thin film growth and structural characterization under the supervision of T.W.N. K.-J.G., J.J., and S.-Y.C. carried out atomic-scale imaging and deep learning analysis. S.Y.P., J.B., and K.M.R. carried out first-principles calculations. C.J.R. and J.S.L. carried out second harmonic generation experiments. J.K. and H.G.L. supported structural characterization. D.L. and J.R.K. wrote the paper with comments from all co-authors. D.L. directed the overall research.

## Competing interests

The authors declare no competing interests.
