## [Peer Review File · Nature Communications]

Reviewers' comments:

Reviewer #1 (Remarks to the Author):

Jeong Rae Kim and collaborators report on the design of perovskite layers with non-equilibrium rotation patterns, inducing functionalities not present in the equilibrium state. As an example they choose the prototypical perovskite CaTiO₃. DFT calculations show that a-a-a- and a+a+c- rotation patterns allow for polar structures with the space groups R3c and P42/mc, respectively, with R3c being the closest in energy. In addition, a large polarization of 44 microcoulombs/cm² is predicted in this metastable material.

To stabilize it, the CaTiO₃ films are grown on (111)- oriented LaAlO₃ substrates, which have the desired R3c symmetry. STEM and PFM measurements proof the rotation pattern, polar distortions and ferroelectric behaviour of the layers.

The results are interesting and deserve to be published. However, more information is needed to properly assess the relevance of the experimental part of the work:

The authors do not describe in detail what is the lattice mismatch between CaTiO₃ and LaAlO₃ in this orientation. This is important because the presence of polarization out of the plane, along the [111] direction, and the associated a-a-a- rotation pattern (they are linked to each other by symmetry) can be achieved by compressively straining the cubic perovskite by the substrate epitaxy. If a long (polar) axis is created in this way, in this orientation, the symmetry will be trigonal (R3c or R3m) and the rotation patterns can only be a-a-a-. Even if the lattice parameters of the two materials are similar, a large number of oxygen vacancies in the CaTiO₃ could provide the necessary strain.

In the manuscript the lattice mismatch, also at the growth temperature, is not clearly discussed. The authors mentioned that calculation show that no polar phase is induced for strains upto 2% but this seems to be in disagreement with some other literature reports (see APL106, 162904 (2015) and DOI:10.1103/PhysRevB.85.064117). It is relevant to mention that CaTiO₃ is reported to be an incipient ferroelectric (see E. Cockayne and B. P. Burton, Phys. Rev. B62, 3735(2000).)

Related to that problem, the authors should explain why the material does not grow polar on the (001) orientation. Since the (001)-LaAlO₃ substrate also has the matching R3c structure, I would expect the growth of metastable R3c CaTiO₃ should also be possible in this orientation, only now with an in-plane component of the polarization.

So, it is not totally clear what is causing the change of symmetry, how the experiments reported here are different from those on strain engineering and how this paper relates to previous reports on the stabilization of polar phases in CaTiO₃ (<https://doi.org/10.1063/1.5078706>). The authors themselves mention that heteroepitaxy is not sufficient for changing the rotation patterns. Then, one wonders if it is the constrain of the rotation pattern what changes the symmetry, giving rise to polar distortions or, on the contrary, it is the strain what creates a polar distortion that induces the change in rotation pattern. If the latter, is this different from strain engineering? The evolution of the symmetry with thickness could give a clue to differentiate these two scenarios.

In my opinion, these issues need to be clarified before the paper can be published in Nat. Comm.

A minor comment is that the Pmna structure is not polar but antipolar: it is not that the cations are positioned at the inversion center, but rather they are displaced in alternating antiparallel fashion.

Reviewer #2 (Remarks to the Author):

Referee report on manuscript NCOMMS-20-09588

In this paper the authors report on the realization of CaTiO₃ films grown on 111 LaAlO₃ substrates. The authors propose that the growth on a substrate with a tilt pattern a-a-a- induces the same tilt pattern in the CaTiO₃ film and stabilizes a ferroelectric state. The paper includes standard film characterization and advanced STEM measurements. Second harmonic generation studies reveal a breaking of inversion symmetry in the 111 CaTiO₃ films. Piezoelectric force microscopy indicates convincingly that the CaTiO₃ films grown on LaNiO₃ buffered 111 LaAlO₃ substrates are ferroelectric. This is a nice and exciting paper in a lively field of research. While the coupling of distortions at oxide interfaces is in general short range, it is proposed here that it can be induced on substantial distances and generate a change in the properties of a material.

As mentioned, this paper is interesting and could be of interest for Nature Communications. I have however a couple of serious concerns that should be addressed before a decision can be made.

Structure and DFT

1. On page 3, the authors mention that their DFT calculations show that the lowest energy structure for CaTiO₃ is the Pnma orthorhombic phase (a-b+a-) – that is the one of bulk CaTiO₃ – a reference to the experimental determination of the structure of CaTiO₃ should be given. In relation with this, I do not understand the sentence p. 4 “The other OOR patterns have mostly not been reported in bulk CaTiO₃”. Are there reports of different ground state structures for CaTiO₃?
2. The authors claim that, in the case of films grown on LaAlO₃ as described here, the [111] direction of growth is essential in stabilizing the ferroelectric state. This claim does indeed seem to be backed up by the data presented in the supplementary figure 10. It should be discussed further however and the paper would benefit to have the comparison between the [111] and [001] growth directions in the main text. Do ab initio calculations taking into account the substrate atoms and strain state predict the (001) interface to promote a non-ferroelectric Pnma and the (111) interface to promote ferroelectric R3c? This is important to address as ferroelectricity has been reported in (001) CaTiO₃ films previously (see for instance, Biegalski et al, Appl. Phys. Lett. 106, 162904 (2015)) and several materials close to ferroelectricity (SrTiO₃ is a prominent example) are easy to push to ferroelectricity by biaxial strain alone.
3. A sketch of the difference between R3c and R-3c would be useful.
4. On page 3, is 100meV per fu a moderate energy cost?

Polarization:

1. The SHG measurements, Fig. 2, reveal a breaking of inversion symmetry. Since these measurements are performed without bottom electrode, what is the direction of the polarization?
2. Still on this point, it is not obvious to understand a polar state in a a-a-a- structure. The paper would benefit from a Figure showing the polar structure and the atomic displacements that generate the polarization.
3. This may also help understand Figure 3e. There, the polarization behaves weirdly. Is the studied film grown on LaNiO₃? If not, it is hard to understand why the polarization would not be in plane? And what is happening at the interface?
4. The 1.3 nm thick film that does not show any signal in SHG, is it R3c, thus the loss of signal is just the ferroelectric size effect? Or could that film be Pnma and therefore not in the correct symmetry to begin with? The system could also choose to develop an in-plane polarization – this makes the answer

to point 2 just above even more important.

STEM

1. It is notoriously difficult to image oxygens by electron microscopy (although the images presented here are beautiful). Could the authors comment on why they could not obtain similar information on the symmetry from ADF images, e.g. looking at antipolar cation motions that should be unique to the Pnma phase or examining the cation off-centering that presumably leads to ferroelectricity in the R3c phase?
2. The symmetry change induced by the electron beam at 200 keV is surprising as such a transition must require a lot of energy and it is not obvious how the electron beam that is focused on a nanometre scale could affect the entire film coherently without inducing significant defects. Do the authors have larger ABF images to show that the new structure is a perfectly coherent Pnma crystal?
3. Along the same lines, the a-b+a- structure must have equivalent geometric domains a+b-b- and a-c+. Are such domains seen?
4. Could the authors subject a R3c film to a similar high energy effect in order to switch the symmetry and then verify by PFM that the film is no longer ferroelectric?

Reviewer #3 (Remarks to the Author):

In this work, the authors exploited strain at hetero-interface to stabilize metastable a-a-a- oxygen octahedral rotation in CaTiO₃ thin films by growing on (111) LaAlO₃, which leads to the polar R3c structure. SHG studies reveal 3-fold in-plane polar symmetry in the 2.6 nm film, with the polar distortion corroborated by ABF-STEM imaging combined with deep neural network analysis. PFM studies show that the polarization is switchable at room temperature, consistent with First-principles calculations. The polar distortion is absent in the 1.3 nm film, which has been attributed to the finite size effect. The authors also showed the metastable OOR can be perturbed by high energy electron beams. Exploiting OOR design to enable novel functionalities in complex oxide is an emerging field with both fundamental interests and technological potential. The current work can thus appeal to a broad audience in the oxide community, and the conclusion is supported by several complementary techniques. However, there are a few technical and presentation deficiencies that need to be properly addressed before I can support its publication in Nature Communications. Below are the detailed comments.

1. Page 4, last paragraph: "Although heteroepitaxy can be used to... cannot be converted into an out-of-phase rotation (e.g., a0b-a0)." It seems the authors were trying to undermine the power of the "heteroepitaxy" approach. This is very strange given that is exactly the technique the authors employed in the current work. Similar statement can also be found in the second paragraph on Page 3 "This has motivated... a significant challenge." Also, the effectiveness of modified OOR pattern on controlling the electronic and magnetic properties of perovskites have been demonstrated in prior works, e.g., inducing polar metal states (Kim et al. Nature 533, 68 (2016)), enhancing magnetic anisotropy (Rajapitamahuni et al., PRL 116, 187201 (2016)), inducing metal-insulator transition (Schütz et al., PRL 119, 256404 (2017)), etc. For an epitaxial thin film, the strain effects on the cation-oxygen bonding length (e.g., Nelson-Cheeseman, Adv Func Mater 24, 6884 (2014)), bonding angle, oxygen octahedral in-plane rotation and out-of-plane tilt are intricately connected, and it seems quite artificial to separate them.
2. The polar axis is not specified. From Fig. 3e, it seems the polar axis is [110], which is consistent with the 3-fold in-plane polar symmetry revealed by the SHG polar plot (Fig. 2d). Does it agree with the theoretical prediction? Also, the equivalent polar axes would lead to domain formation. In Fig. 4,

the authors showed the vertical PFM images of domain structures written with applied voltage. It'd be informative to show what the pristine state of the sample is without external field applied. Is there out-of-plane domain observed? I also suggest the authors perform lateral PFM to map out the in-plane polarization distribution.

3. It is known that epitaxial strain relaxes with film thickness. Have the authors examined thicker films to explore the critical thickness where the enforced polar distortion disappears?

4. Page 7, second paragraph: "Bipolar domain patterns (Fig. 4c,d), created in CaTiO₃ (111) films, also remain stable for at least several hours." Please avoid such vague description and give the details of time-dependence PFM experiments, i.e., the time duration, time-dependence of the signal strength, etc. Also, the authors should note that there are relaxor systems where the PFM response for the written domains can last for days, e.g., Moghadam et al., Nano Lett., 17, 6248 (2017).

Responses to the comments of Reviewer #1

We would like to thank the reviewer for his/her in-depth review and excellent questions/suggestions regarding our manuscript. In the pages that follow, we provide our responses to each of the reviewer's comments, in order. The responses are written in blue.

Jeong Rae Kim and collaborators report on the design of perovskite layers with non-equilibrium rotation patterns, inducing functionalities not present in the equilibrium state. As an example they choose the prototypical perovskite CaTiO₃. DFT calculations show that $a^-a^-a^-$ and $a^+a^+c^-$ rotation patterns allow for polar structures with the space groups R3c and P42/mc, respectively, with R3c being the closest in energy. In addition, a large polarization of 44 microcoulombs/cm² is predicted in this metastable material.

To stabilize it, the CaTiO₃ films are grown on (111)- oriented LaAlO₃ substrates, which have the desired R3c symmetry. STEM and PFM measurements proof the rotation pattern, polar distortions and ferroelectric behaviour of the layers.

The results are interesting and deserve to be published. However, more information is needed to properly assess the relevance of the experimental part of the work:

We thank the reviewer for positive evaluation of our study.

Question#1

The authors do not describe in detail what is the lattice mismatch between CaTiO₃ and LaAlO₃ in this orientation.

Response#1

We thank the reviewer for the comment. The reviewer is correct in pointing out that we did not provide important information on the lattice mismatch. In the revised Supplementary Information, we comprehensively present the lattice mismatch between LaAlO₃ and CaTiO₃, based on DFT calculations. Since the $a^-a^-a^-$ pattern does not exist in bulk, we consistently compare theoretically calculated lattice parameters of LaAlO₃, $Pnma$ -CaTiO₃, $R\bar{3}c$ -CaTiO₃, and $R3c$ -CaTiO₃ (Fig. R1 & Table R1). Due to the distinct structural symmetries of $a^-a^-a^-$ and $a^-b^+a^-$ OOR, explicit calculation of epitaxial strain applied to the $Pnma$ -CaTiO₃ film on LaAlO₃ substrate is not available. So, we estimated the strain using the volume-conserving cubic lattice constants of CaTiO₃ [$a = (V_{Pnma}/4)^{1/3} = 3.773$ Å] and LaAlO₃ [$a = (\sqrt{3} a_H^2 \times c_H / 12)^{1/3} = 3.756$ Å]. For $R\bar{3}c$ -CaTiO₃ and $R3c$ -CaTiO₃, explicit calculation of epitaxial

strain on LaAlO_3 (111)_{pc} substrate is available. Overall, CaTiO_3 films are in small compressive strain below 1 %. We added these data in the revised Supplementary Information as Supplementary Fig. S2 and Table S1.

Fig. R1 | DFT calculation of structures of CaTiO_3 . a,b,c, DFT calculation of CaTiO_3 with $Pnma$ (a), $R\bar{3}c$ (b), and $R3c$ (c) structures.

Type	Hexagonal		Orthorhombic			$(Vol./f.u.)^{1/3}$
	a	c	a	b	c	a
LaAlO_3	5.315	12.994	N/A	N/A	N/A	3.756
$Pnma\text{-CaTiO}_3$	N/A	N/A	5.397	7.527	5.288	3.773
Strain (%)	N/A	N/A	N/A	N/A	N/A	-0.45
$R\bar{3}c\text{-CaTiO}_3$	5.370	12.990	N/A	N/A	N/A	3.781
Strain (%)	-1.02	0.03	N/A	N/A	N/A	-0.67
$R3c\text{-CaTiO}_3$	5.360	13.133	N/A	N/A	N/A	3.791
Strain (%)	-0.84	-1.06	N/A	N/A	N/A	-0.91

Table R1 | Table of lattice parameters and strain values from DFT

Question#2

This is important because the presence of polarization out of the plane, along the [111] direction, and the associated a-a-a- rotation pattern (they are linked to each other by symmetry) can be achieved by compressively straining the cubic perovskite by the substrate epitaxy. If a long (polar) axis is created in this way, in this orientation, the symmetry will be trigonal ($R\bar{3}c$ or $R3m$) and the rotation patterns can only be a-a-a-.

Response#2

We agree with the reviewer's comment. When grown "fully coherently" on LaAlO_3 (111) substrate, CaTiO_3 can only show the $a^-a^-a^-$ pattern. In fact, we already emphasized this in our original manuscript, by stating "Figures 2a,b emphasize that on the (111) plane, only the $a^-a^-a^-$ pattern allows for a regular hexagon network of A-site ions.". As will be discussed in detail in Response#8, however, our theoretical and experimental works consistently reveal that

the compressive strain itself is not sufficient for achieving ferroelectricity in CaTiO₃ (111) films. The total energy of the strained *R3c* structure is always higher than that of strained *Pnma* structure in the reasonable strain range (from -2% to 2%) as discussed in Fig. S4. Therefore, only when CaTiO₃ (111) films are geometrically constrained under the regular hexagon (or regular triangle) atomic network [i.e., coherently to the in-plane lattice symmetry of LaAlO₃ (111)], they could exhibit the $a^-a^-a^-$ pattern and resulting ferroelectricity.

Question#3

Even if the lattice parameters of the two materials are similar, a large number of oxygen vacancies in the CaTiO₃ could provide the necessary strain.

Response#3

We thank the reviewer for the important comment. We confirmed that when grown under our optimized condition, CaTiO₃ films possess excellent crystallinity (see, e.g., Supplementary Fig. S7); their lattice parameters were the same as those of CaTiO₃ grown by molecular beam epitaxy technique [R. C. Haislmaier *et al.*, *Adv. Funct. Mater.* **26**, 7271 (2016)], suggesting good chemical stoichiometry of our CaTiO₃ films. Thus, we believe that only a small number of oxygen vacancies are present in our films, which cannot affect our arguments.

Question#4

In the manuscript the lattice mismatch, also at the growth temperature, is not clearly discussed.

Response#4

In Response#1, we provided comprehensive information on the lattice mismatch in Fig. R1, Table. R1. We also added these data in the revised Supplementary Information as Supplementary Fig. S2 and Table S1.

To calculate the experimental lattice mismatch at room temperature and the growth temperature (873 K), we did literature search [C. J. Howard *et al.*, *J. Phys.: Condens. Matter* **12**, 249 (2000), M. Yashima, R. Ali, *Solid State Ionics* **180**, 120 (2009)]. At room temperature, LaAlO₃ (295 K) is rhombohedral with $a = b = 5.3646 \text{ \AA}$, $c = 13.1095 \text{ \AA}$ and CaTiO₃ (296 K) is orthorhombic with $a = 5.3709 \text{ \AA}$, $b = 5.4280 \text{ \AA}$, $c = 7.6268 \text{ \AA}$. The volume-conserving cubic lattice constants are $a = b = c = 3.7904 \text{ \AA}$ for LaAlO₃ and $a = b = c = 3.8164 \text{ \AA}$ for CaTiO₃, resulting in lattice mismatch of -0.6877%. Near the growth temperature, LaAlO₃ (848 K) becomes cubic with $a = b = c = 3.8116 \text{ \AA}$ and CaTiO₃ (868 K) is still orthorhombic with $a = 5.4179 \text{ \AA}$, $b = 5.4478 \text{ \AA}$, $c = 7.6834 \text{ \AA}$. The volume-conserving cubic lattice constants are $a = b = c = 3.8116 \text{ \AA}$ for LaAlO₃ and $a = b = c = 3.8416 \text{ \AA}$ for CaTiO₃, resulting in lattice mismatch of -0.7878%. After comparing the two cases (i.e., at room temperature and growth temperature), we checked that the lattice mismatch is consistently below 1% and concluded that the temperature dependence is minute.

Question#5

The authors mentioned that calculation show that no polar phase is induced for strains upto 2% but this seems to be in disagreement with some other literature reports (see APL106, 162904 (2015) and DOI:10.1103/PhysRevB.85.064117). It is relevant to mention that CaTiO₃ is reported to be an incipient ferroelectric (see E. Cockayne and B. P. Burton, Phys. Rev. B62, 3735(2000).)

Response#5

Please note that our calculations (Supplementary Fig. S4) considered strain dependence of CaTiO₃ (111) films, whereas the other reports mentioned by the reviewer dealt with the case of CaTiO₃ (001) films. Despite the different film directions, however, our calculation results are somehow consistent with the previous reports. The previous reports found that tensile-strained CaTiO₃ (001) films could become ferroelectric more effectively than compressive-strained CaTiO₃ (001). Similarly, our calculations found that some phonons (e.g., Y point) become more softened in the +2 % tensile-strained CaTiO₃ (111) than in the -2 % compressive-strained CaTiO₃ (111); it might thus be possible that polar phases emerge under extremely high tensile strains in CaTiO₃ (111) films in our calculation as well. However, such a case would not be relevant to our experimental conditions.

Following the reviewer's comment, we added the statement that CaTiO₃ is an incipient ferroelectric material (Line #61 of our revised manuscript).

Question#6

Related to that problem, the authors should explain why the material does not grow polar on the (001) orientation. Since the (001)-LaAlO₃ substrate also has the matching R3c structure, I would expect the growth of metastable R3c CaTiO₃ should also be possible in this orientation, only now with an in-plane component of the polarization.

Response#6

We agree with the reviewer that, if the octahedral connectivity is maintained at the interface, the both orientations might result in the R3c structures. However, our experimental findings show the orientation-dependent stabilization of the rotation pattern. This suggests that there exists a difference in the octahedral connectivity between the (001) and (111) surfaces. In Fig. R2, we illustrate the Ti-O-Al network, where the (001) interface has one oxygen atom connecting two layers and the (111) interface has three oxygen atoms connecting two layers. Converting it to the density of Ti-O-Al bond per unit area, the values for the (001) and (111) interfaces are about $1/a_c^2$ and $\sqrt{3}/a_c^2$, respectively. Thus, the (111) interface has about 1.7 times larger bond density than (001) interface. This robust connectivity strongly locks the interfacial OOR (Fig. R2b); at the (111) interface, the OOR pattern of CaTiO₃ would more

likely follow that of LaAlO_3 .

Furthermore, as explained in Response#2, the unique geometric constraint (i.e., regular hexagon atomic network) of the (111) interface could selectively stabilize the $R3c$ CaTiO_3 . On the other hand, the (001) interface is unlikely to selectively stabilize the $R3c$ phase. We added this discussion in the revised Supplementary Information as Supplementary Fig. S5.

Fig. R2 | Octahedral connectivity across $(001)_{pc}$ and $(111)_{pc}$ heterointerface. a, Octahedral connectivity across $(001)_{pc}$ -oriented heterointerface. One bond connection allows the interfacial OOR relationships to be either in-phase or out-of-phase. **b,** Octahedral connectivity across $(111)_{pc}$ -oriented heterointerface. Three bond connections lock the interfacial OOR pattern configurations.

Question#7

So, it is not totally clear what is causing the change of symmetry, how the experiments reported here are different from those on strain engineering and how this paper relates to previous reports on the stabilization of polar phases in CaTiO_3 (<https://doi.org/10.1063/1.5078706>).

Response#7

As explained in our Response#6, the geometric constraint and robust octahedral connectivity at the $\text{CaTiO}_3/\text{LaAlO}_3$ (111) interface could cause the change of symmetry in CaTiO_3 . In the next Response#8, we will clarify why the epitaxial strain itself cannot be the primary origin of our observation, i.e., why our approach is distinct from strain engineering.

In the article mentioned by the reviewer, the polar phase is the exotic surface states of CaTiO_3 . On the other hand, our work demonstrates that an intrinsic polar phase emerges in the bulk of CaTiO_3 (111) films.

Question#8

The authors themselves mention that heteroepitaxy is not sufficient for changing the rotation patterns. Then, one wonders if it is the constrain of the rotation pattern what changes the symmetry, giving rise to polar distortions or, on the contrary, it is the strain what creates a polar distortion that induces the change in rotation pattern. If the latter, is this different from strain engineering? The evolution of the symmetry with thickness could give a clue to differentiate these two scenarios.

Response#8

We thank the reviewer for raising this critical issue. Figure 3 in the main text reveals that when imaged with a higher electron kinetic energy (e.g., 200 keV), the OOR pattern relaxes from a^-a^- (i.e., polar $R3c$) to $a^-b^+a^-$ [i.e., nonpolar (or antipolar) $Pnma$]. Importantly, Fig. R3 below shows that even after this a^-a^- -to- $a^-b^+a^-$ relaxation, CaTiO_3 is still fully strained to the underlying LaAlO_3 (111) substrate, that is, the in-plane lattice of CaTiO_3 remains identical to that of LaAlO_3 (111). This emphasizes that the epitaxial strain itself can stabilize neither ferroelectricity nor a^-a^- OOR pattern in CaTiO_3 films on LaAlO_3 (111) substrate. This is also consistent with our DFT calculations (Supplementary Fig. S4), indicating that the epitaxial strain itself (up to $\pm 2\%$) is not the primary origin for the ferroelectricity in CaTiO_3 (111) films. We added this data in the revised Supplementary Information as Supplementary Fig. S14.

Fig. R3 | CaTiO_3 in-plane lattice before and after the OOR-pattern relaxation.

Furthermore, following the reviewer's suggestion, we conducted additional experiments on the evolution of the symmetry with thicknesses. Please note that, thanks to the small lattice mismatch between CaTiO_3 and LaAlO_3 (Table. R1), all CaTiO_3 films considered were fully strained on the LaAlO_3 (111) substrates. Reciprocal space mapping carried out on representative 25 nm-thick CaTiO_3 is given in Fig. R4 below.

We carried out ABF-STEM analysis on 5 nm-, 20 nm-, and 36 nm-thick CaTiO₃/LaAlO₃ (111) films. As with 2.6 nm-thick CaTiO₃ (Fig. 3), 5 nm-thick CaTiO₃ shows the single phase of *R3c* (Fig. R5 below). However, a distinct behavior is seen from 20 nm- (Fig. R6) and 36 nm-thick films (Fig. R7). Atomic-scale analysis of the OOR pattern was performed on several regions with varying distances from the CaTiO₃/LaAlO₃ interface. For both films, we consistently observed that, up to 7 nm from the interface, the OOR pattern was $\bar{a}^- \bar{a}^- \bar{a}^-$ (i.e., polar *R3c*). However, in distant regions from the interface (e.g., above 8 nm), the OOR pattern began relaxed to $a^- b^+ a^-$ [i.e., nonpolar (or antipolar) *Pnma*]. Considering the competition between the bulk energy (preferring the *Pnma* phase) and interfacial interaction (preferring the *R3c* phase), this thickness dependence is quite reasonable: our data suggest that for thicknesses above around 8 nm, the bulk energy contribution starts to prevail over the interfacial interaction contribution. This observation repeatedly demonstrates that epitaxial strain itself is not the primary origin of either ferroelectricity or OOR pattern change in CaTiO₃ (111) films.

Fig. R4 | Fully-strained CaTiO₃/LaAlO₃ (111) thin film. Reciprocal space mapping of 25 nm-thick CaTiO₃/LaAlO₃ (111)_{pc} thin film near LaAlO₃ (312)_{pc} peak. CaTiO₃ (312)_{pc} and LaAlO₃ (312)_{pc} are peaked at an identical H value, meaning identical in-plane lattice parameters.

[Redacted]

[Redacted]

[Redacted]

Question#9

A minor comment is that the Pmna structure is not polar but antipolar: it is not that the cations are positioned at the inversion center, but rather they are displaced in alternating antiparallel fashion.

Response#9

We thank the reviewer for the explanation and have accordingly mentioned it in our revised manuscript (Line #60).

Responses to the comments of Reviewer #2

We would like to thank the reviewer for his/her in-depth review and excellent questions/suggestions regarding our manuscript. In the pages that follow, we provide our responses to each of the reviewer's comments, in order. The responses are written in blue.

In this paper the authors report on the realization of CaTiO₃ films grown on 111 LaAlO₃ substrates. The authors propose that the growth on a substrate with a tilt pattern $a-a-a$ induces the same tilt pattern in the CaTiO₃ film and stabilizes a ferroelectric state. The paper includes standard film characterization and advanced STEM measurements. Second harmonic generation studies reveal a breaking of inversion symmetry in the 111 CaTiO₃ films. Piezoelectric force microscopy indicates convincingly that the CaTiO₃ films grown on LaNiO₃ buffered 111 LaAlO₃ substrates are ferroelectric.

This is a nice and exciting paper in a lively field of research. While the coupling of distortions at oxide interfaces is in general short range, it is proposed here that it can be induced on substantial distances and generate a change in the properties of a material. As mentioned, this paper is interesting and could be of interest for Nature Communications. I have however a couple of serious concerns that should be addressed before a decision can be made.

We thank the reviewer for positive evaluation of our work.

Question#1

Structure and DFT

1. On page 3, the authors mention that their DFT calculations show that the lowest energy structure for CaTiO₃ is the *Pnma* orthorhombic phase ($a-b+a$) – that is the one of bulk CaTiO₃ – a reference to the experimental determination of the structure of CaTiO₃ should be given.

In relation with this, I do not understand the sentence p. 4 “The other OOR patterns have mostly not been reported in bulk CaTiO₃”. Are there reports of different ground state structures for CaTiO₃?

Response#1

We thank the reviewer for pointing out our mistake in addressing previous works on CaTiO₃. First, we added a reference for the *Pnma* structure of CaTiO₃ [M. Yashima, R. Ali, *Solid State Ionics* **180**, 120-126 (2009)]. Regarding the second point, we admit that the statement is confusing. We noticed that unusual monoclinic *Pc* phase has been reported in CaTiO₃ nanoparticles and surface [M. O. Ramirez, *et al.*, *APL Mater.* **7**, 011103 (2019)]. This cannot have the $a^-b^+a^-$ OOR pattern, so we used such vague statements in our manuscript. In the revised manuscript, we removed the confusing statement.

Question#2

2. The authors claim that, in the case of films grown on LaAlO₃ as described here, the [111] direction of growth is essential in stabilizing the ferroelectric state. This claim does indeed seem to be backed up by the data presented in the supplementary figure 10. It should be discussed further however and the paper would benefit to have the comparison between the [111] and [001] growth directions in the main text.

Response#2

We appreciate the reviewer's valuable comments. In the revised manuscript, we provided further comparative discussions on [111] and [001] growth directions in the Line #161 of the revised manuscript and included the PFM data of [001]-oriented CaTiO₃ in the main Fig. 4.

Question#3

Do *ab initio* calculations taking into account the substrate atoms and strain state predict the (001) interface to promote a non-ferroelectric *Pnma* and the (111) interface to promote ferroelectric *R3c*? This is important to address as ferroelectricity has been reported in (001) CaTiO₃ films previously (see for instance, Biegalski et al, Appl. Phys. Lett. 106, 162904 (2015)) and several materials close to ferroelectricity (SrTiO₃ is a prominent example) are easy to push to ferroelectricity by biaxial strain alone.

Response#3

In our *ab-initio* calculations, we have considered the effect of strain but have not included the substrate atoms. The effect of substrate atoms could be included by using a large supercell with more than 500 atoms depending on the atomic relaxation length and the type of interface, which is challenging to calculate due to large computational cost. More importantly, since the growth process involves many high-energy processes, even with such calculation results, the obtained information (e.g., the ground-state energy differences among different orientations) is not sufficient to determine which rotation pattern is preferred. Especially, the energy difference between the *Pnma* and *R3c* structures (~ 75 meV f.u.⁻¹) and observed switching of the rotation patterns from $a^-a^-a^-$ to $a^-b^+a^-$ by electron beam strongly suggest that the $a^-a^-a^-$ rotation pattern is not globally stable from the interfacial interaction but locally stable induced by growth kinetics. This is also consistent with our calculation results showing that the *R3c* structure is locally stable. In this case, obtaining the ground-state energy of the supercell including substrate atoms, in our opinion, would not support the interface-dependent rotation pattern. The effect of strain is included by fixing the in-plane lattice constants during the atomic relaxation with a reasonable range of strain values (-2% to $+2\%$). For [001]-oriented CaTiO₃ film, as the reviewer pointed out, there is polar instability reported for the tensile strained films and not for the compressively strained films. Since our [001]-oriented film on LaAlO₃ is under small compressive strain, we do not expect the polar instability in our [001]-oriented film. For

[111]-oriented films under strains from -2% to $+2\%$ (Supplementary Fig. S4), *Pnma* was always more stable than *R3c*. Furthermore, the energy difference increases under compressive strains; compressive strain makes *Pnma* more stable compared to *R3c*. So, our observation is not a result of epitaxial strain, but interfacial geometric constraint (Fig. 2) and octahedral connectivity (Fig. R2).

Question#4

3. A sketch of the difference between *R3c* and *R-3c* would be useful.

Response#4

We agree that providing further structural information on *R3c* and $\bar{R}3c$ would help readers understanding. As recommended by the reviewer, we added visualized atomic structures and table of lattice parameters of *R3c* and $\bar{R}3c$ in the Supplementary Information as Supplementary Fig. S2 and Table S1.

Question#5

4. On page 3, is 100meV per fu a moderate energy cost?

Question#5

We searched for earlier works on heteroepitaxial stabilization of a nonequilibrium phase. We found that HoMnO_3 is hexagonal in bulk but can be stabilized as an orthorhombic phase in thin film [T. H. Lin *et al.*, *Appl. Phys. Lett.* **92**, 132503 (2008)]. DFT calculations find that hexagonal and orthorhombic HoMnO_3 show the total energy difference of 71–156 meV f.u.⁻¹ [C.-Y Ren, *Phys. Rev. B* **79**, 125113 (2009)]. In our case, the *R3c* CaTiO_3 phase has the energy cost of ~ 75 meV f.u.⁻¹. Therefore, we believe that the energy cost of <100 meV f.u.⁻¹ could be moderate enough to be overcome by the [111]-directed film growth.

Question#6

Polarization:

1. The SHG measurements, Fig. 2, reveal a breaking of inversion symmetry. Since these measurements are performed without bottom electrode, what is the direction of the polarization?

Response#6

The SHG we observed in $\text{CaTiO}_3/\text{LaAlO}_3$ (111) heterostructure is qualitatively same with the results from (111)-oriented LiNbO_3 single crystal with out-of-plane polarization [T. J. Sono, *et al.*, *Phys. Rev. B* **74**, 205424 (2006)]. Considering that LiNbO_3 is also in the *R3c* space group, the results of Fig. 2d is seemingly coming from the domain #1 of *R3c*- CaTiO_3 with out-of-plane polarization (Fig. R8 below).

Fig. R8 | Four possible domains of $R3c$ CaTiO_3 stabilized on LaAlO_3 (111) substrate. Black arrows indicate the Ti polar displacement.

Question#7

2. Still on this point, it is not obvious to understand a polar state in a a-a-a- structure. The paper would benefit from a Figure showing the polar structure and the atomic displacements that generate the polarization.

Response#7

We thank the reviewer for pointing out a lack of information in our manuscript. To provide comprehensive information on the polar state of $R3c$ - CaTiO_3 , we added a visualized atomic structure and polar domain structure of $R3c$ - CaTiO_3 (Supplementary Figs. S2 and S16).

Question#8

3. This may also help understand Figure 3e. There, the polarization behaves weirdly. Is the studied film grown on LaNiO_3 ? If not, it is hard to understand why the polarization would not be in plane? And what is happening at the interface?

Response#8

Figure 3e shows the polarization map, corresponding to the domain #4 (as depicted in Fig. R8) of $R3c$ CaTiO_3 grown directly on LaAlO_3 (111) substrate, without the LaNiO_3 buffer layer. [Please note that in the revised Supplementary Information, we provided the polarization map images for all four possible domains of $R3c$ CaTiO_3 (Supplementary Fig. S16).] It seems that the reviewer is concerned about the depolarization field in ultrathin ferroelectrics, under which the out-of-plane polarization may not be preferred. In $R3c$ CaTiO_3 , however, the polar axis is determined as the rhombohedral c axis; thus, if present, polarization should have the out-of-plane component in CaTiO_3 (111) films. In addition, a slight compressive strain, as well as the charge discontinuity at the interface between $\text{Ca}^{2+}\text{Ti}^{4+}\text{O}^{2-}_3$ and $\text{La}^{3+}\text{Al}^{3+}\text{O}^{2-}_3$, could also compensate for the effect from the depolarization field.

Also, the reviewer seems to be concerned about the weird behavior of polarization (i.e., suppressed polarization) near the interface. To address this concern, we carried out DFT

calculation of realistic heterostructures. A superlattice consisting of 12 layer LaNiO_3 and 12 layer CaTiO_3 , $[(\text{LaNiO}_3)_{12}/(\text{CaTiO}_3)_{12}]$ was considered in our DFT calculation. When we did DFT calculation of $[(\text{LaAlO}_3)_{12}/(\text{CaTiO}_3)_{12}]$ superlattice, we found that interfacial charge discontinuity dope charge carriers to the CaTiO_3 , which should be avoided [H. Chen, A. Kolpak, and S. Ismail-Beigi, *Phys. Rev. B* **82**, 085430 (2010)]. In the case of $[(\text{LaNiO}_3)_{12}/(\text{CaTiO}_3)_{12}]$ superlattice, electrons in LaNiO_3 effectively screened the charge discontinuity, and we think replacing LaAlO_3 with LaNiO_3 would not affect our arguments to be discussed below. Figure R9 below shows the calculated structures of $[(\text{LaNiO}_3)_{12}/(\text{CaTiO}_3)_{12}]$ superlattices and corresponding layer-resolved ionic displacements. We observed that, independent of interfacial terminations, interface dipoles are always pointing from CaTiO_3 to LaNiO_3 . These interface dipoles suppress (or enhance) the originally rightward, intrinsic polarization of $R3c$ - CaTiO_3 near the 11th or 12th layers (or near the 23th and 24th layers). These results are consistent with our experimental observation in Fig. 3e. Therefore, we conclude that the weird behavior of interfacial polarization could be due to interface dipoles at the $\text{CaTiO}_3/\text{LaAlO}_3$ (111) interface.

Fig. R9 | DFT calculation of interface electric dipole at $\text{LaNiO}_3/\text{CaTiO}_3$ interface. a,b, Schematic of $[(\text{LaNiO}_3)_{12}/(\text{CaTiO}_3)_{12}]$ superlattices for DFT calculation with Ni/CaO₃ (a) and LaO₃/Ti (b) interface terminations, respectively. **c,d,** DFT calculation of layer-resolved ionic displacement in the Ni/CaO₃-terminated (c) and LaO₃/Ti-terminated (d) $[(\text{LaNiO}_3)_{12}/(\text{CaTiO}_3)_{12}]$ superlattices.

Question#9

4. The 1.3 nm thick film that does not show any signal in SHG, is it $R3c$, thus the loss of signal is just the ferroelectric size effect? Or could that film be $Pnma$ and therefore not in the correct symmetry to begin with? The system could also choose to develop an in-plane polarization – this makes the answer to point 2 just above even more important.

Response#9

It is an interesting point to consider. Unfortunately, however, structural characterization on the 1.3 nm-thick films was extremely challenging due to experimental limitations. In our opinion, the interfacial effect [i.e., geometric constraint (Fig. 2) and octahedral connectivity (Fig. R2)] would be maximized in the ultrathin limit of CaTiO_3 (111) film; we thus expect that the OOR pattern of 1.3 nm-thick CaTiO_3 (111) films is still $a^-a^-a^-$. If any polarization (including the in-plane one) is present, then the SHG signal must be present as well. Therefore, the structure of 1.3 nm-thick CaTiO_3 (111) film should be nonpolar $R\bar{3}c$.

Question#10

STEM

1. It is notoriously difficult to image oxygens by electron microscopy (although the images presented here are beautiful). Could the authors comment on why they could not obtain similar information on the symmetry from ADF images, e.g. looking at antipolar cation motions that should be unique to the $Pnma$ phase or examining the cation off-centering that presumably leads to ferroelectricity in the $R3c$ phase?

Response#10

The main purpose of our study is to directly image the controlled oxygen octahedral rotation (OOR) pattern. While we calculated the polarization using ADF, we had to utilize ABF images for determining the OOR patterns. As pointed out by the reviewer, the antipolar mode might be unique to the $Pnma$ symmetry in bulk. However, although the absence of antipolar mode of A-site cations may exclude the $Pnma$ symmetry, it does not always indicate the $R3c$ symmetry. For example, $a^0a^0a^0$ trivially does not have the antipolar modes of A-site cations. Furthermore, in some zone axes, the antipolar mode could be hidden even in the $Pnma$ case, as shown in Fig. R10 below. Therefore, we believe that precise determination of the OOR pattern requires the measurement of oxygen position via ABF imaging. Only when the elongated oxygen peaks must be aligned in the same crystalline direction without any zig-zag pattern, we can precisely assign the OOR pattern as $a^-a^-a^-$.

Fig. R10 | Observing antipolar cation motions of *Pnma* structure in STEM. Crystal structure of *Pnma*-CaTiO₃ along the two different zone axes. Depending on the zone axes, the antipolar cation motions might not be observable in STEM.

Question#11

2. The symmetry change induced by the electron beam at 200 keV is surprising as such a transition must require a lot of energy and it is not obvious how the electron beam that is focused on a nanometre scale could affect the entire film coherently without inducing significant defects. Do the authors have larger ABF images to show that the new structure is a perfectly coherent *Pnma* crystal?

Response#11

We appreciate the reviewer's comment. According to our calculations (Fig. 1f), the threshold energy for the *R3c*-to-*Pnma* relaxation is around 10 meV f.u.⁻¹; we thus believe that this relaxation could happen under exposure of high-energy electron beam, and once relaxed, the *Pnma* structure will remain stable, since it is the ground state. Also, our calculations (Fig. 1f) implies the presence of smooth *R3c*-to-*Pnma* transition path (with small threshold energy), which might allow for the transition without significant defects being induced. However, unveiling the exact transition mechanism is beyond the scope of our present study, but it would be an interesting research topic in the future.

Regarding the larger ABF images, the largest ABF image was found to be only about 7.3 nm * 3.6 nm as shown in Fig. R11, because the lowest magnification of STEM is limited up to about 20~25 million times for the verification of oxygen structures. We could see the perfectly coherent structure without the generation of defects in this field-of-view image. For the clarity of relaxation, we put polarization mapping together with images.

Fig. R11 | ABF-STEM imaging of $Pnma$ - CaTiO_3 on LaAlO_3 (111)_{pc} in a large scale. ABF-STEM imaging of $Pnma$ - CaTiO_3 on LaAlO_3 (111)_{pc} at a large scale using electron acceleration voltage of 200 kV. For the observable regions, $Pnma$ - CaTiO_3 is perfectly coherent on LaAlO_3 (111) substrate.

Question#12

3. Along the same lines, the $a^-b^+a^-$ structure must have equivalent geometric domains $a^+b^-b^-$ and $a^-a^-c^+$. Are such domains seen?

Response#12

As correctly expected by the reviewer, we observed the other domains, as shown in Fig. R12 below. Only one kind of atomic structure can be possible along the [110] zone axis both in the cases of $a^+b^-b^-$ and $a^-a^-c^+$. Even if we didn't observe domain wall because of the experimental limitation mentioned above, we found the other domain structures corresponding to $a^+b^-b^-$ and $a^-a^-c^+$. For each domain, we checked the disappearance of electric polarization.

Fig. R12 | Domain structures of $Pnma$ - CaTiO_3 on LaAlO_3 (111) substrate. **a**, Schematic of three possible domain structures of $Pnma$ - CaTiO_3 thin films on (111)-plane of LaAlO_3 . **b**, Atomic scale imaging and polarization mapping on three domain structures of $Pnma$ - CaTiO_3 via ABF-STEM.

Question#13

4. Could the authors subject a R3c film to a similar high energy effect in order to switch the symmetry and then verify by PFM that the film is no longer ferroelectric?

Response#13

This is a very interesting question and comment, implying another pathway to control the metastable phase of CaTiO₃. For the real-time verification by PFM, we searched for a similar high energy effect available in our scanning probe microscopy set-up. We tried to use loading force by a nano-scale probe to apply the high energy effect. However, what we observed is not structural relaxation from *R3c* to *Pnma*, but flexoelectricity-induced mechanical switching of ferroelectric polarization [H. Lu *et al.*, *Science* **336**, 59 (2012)]. At this moment, we would like to further study the suggested idea and continue as independent research.

Fig. R13 | Mechanical control of ferroelectricity in *R3c*-CaTiO₃. Application of gradually increasing tip loading force on bipolar domains written on *R3c*-CaTiO₃ thin films. Ferroelectric polarization is switched by mechanical force.

Responses to the comments of Reviewer #3

We would like to thank the reviewer for his/her in-depth review and excellent questions/suggestions regarding our manuscript. In the pages that follow, we provide our responses to each of the reviewer's comments, in order. The responses are written in blue.

In this work, the authors exploited strain at hetero-interface to stabilize metastable a-a-a-oxygen octahedral rotation in CaTiO₃ thin films by growing on (111) LaAlO₃, which leads to the polar R3c structure. SHG studies reveal 3-fold in-plane polar symmetry in the 2.6 nm film, with the polar distortion corroborated by ABF-STEM imaging combined with deep neutral network analysis. PFM studies show that the polarization is switchable at room temperature, consistent with First-principles calculations. The polar distortion is absent in the 1.3 nm film, which has been attributed to the finite size effect. The authors also showed the metastable OOR can be perturbed by high energy electron beams. Exploiting OOR design to enable novel functionalities in complex oxide is an emerging field with both fundamental interests and technological potential. The current work can thus appeal to a broad audience in the oxide community, and the conclusion is supported by several complementary techniques. However, there are a few technical and presentation deficiencies that need to be properly addressed before I can support its publication in Nature Communications. Below are the detailed comments.

We thank the reviewer for positive evaluation of our study.

Question#1

1. Page 4, last paragraph: "Although heteroepitaxy can be used to... cannot be converted into an out-of-phase rotation (e.g., a0b-a0)." It seems the authors were trying to undermine the power of the "heteroepitaxy" approach. This is very strange given that is exactly the technique the authors employed in the current work.

Similar statement can also be found in the second paragraph on Page 3 "This has motivated... a significant challenge."

Also, the effectiveness of modified OOR pattern on controlling the electronic and magnetic properties of perovskites have been demonstrated in prior works, e.g., inducing polar metal states (Kim et al. Nature 533, 68 (2016), enhancing magnetic anisotropy (Rajapitamahuni et al., PRL 116, 187201 (2016)), inducing metal-insulator transition (Schütz et al., PRL 119, 256404 (2017)), etc. For an epitaxial thin film, the strain effects on the cation-oxygen bonding length (e.g., Nelson-Cheeseman, Adv Func Mater 24, 6884 (2014)), bonding angle, oxygen octahedral in-plane rotation and out-of-plane tilt are intricately connected, and it seems quite

artificial to separate them.

Response#1

We thank the reviewer for the critical comments. We admit that we made a mistake in the statements. As correctly pointed out by the reviewer, we employed the heteroepitaxy in our work, so we did not intend to undermine the power of the heteroepitaxy approach. To avoid any further confusion, we revised sentences in the main text as below:

“Despite extensive works, however, there is still a lack of studies on engineering the pattern itself of OOR and then generating novel functionalities.”

“Through simply modulating OOR angles, it might be difficult to convert the OOR pattern itself from $a^-b^+a^-$ (i.e., corresponding to $Pnma$) to $a^-a^-a^-$ (i.e., corresponding to $R3c$).”

We also admit that we rather overemphasized our novelty on the heteroepitaxial control of OOR. But, we still believe that we have our own novelty in terms of the effectiveness of OOR-pattern control, which was pointed out by Reviewer #2 as well (*Reviewer #2’s comment: While the coupling of distortions at oxide interfaces is in general short range, it is proposed here that it can be induced on substantial distances and generate a change in the properties of a material.*). After carefully considering the prior works [T. H. Kim *et al.*, *Nature* **533**, 68-72 (2016), A. Rajapitamahuni *et al.*, *Phys. Rev. Lett.* **116**, 187201 (2016), P. Schütz *et al.*, *Phys. Rev. Lett.* **119**, 256404 (2017)], we claim that our method can realize the OOR pattern, (i) which is not in group-subgroup relation with the original OOR pattern, (ii) at a substantial length scale. Previous engineering of $a^-b^+a^-$ pattern results in the angle change within the same $a^-b^+a^-$ pattern, or transition to lower-symmetric $a^-b^+c^-$ pattern, a subgroup of $Pnma$ [Q. He, *et al.*, *ACS Nano* **9**, 8412-8419 (2015)]. On the other hand, $a^-a^-a^-$ has totally different structural symmetry (e.g., 3-fold rotational symmetry), which has not been achieved yet. We believe that this is a distinguished feature of our OOR control.

Question#2

2. The polar axis is not specified. From Fig. 3e, it seems the polar axis is [110], which is consistent with the 3-fold in-plane polar symmetry revealed by the SHG polar plot (Fig. 2d). Does it agree with the theoretical prediction?

Response#2

In the $R3c$ structure, the polar axis is determined as the rhombohedral c axis. Since the $R3c$ CaTiO_3 has four kinds of possible domains in (111) films (as shown in Fig. R14a below), the measured polar direction should depend on the domain. In this paper, we put the domain #4 structure in the main Fig. 3. This domain shows both the out-of-plane and in-plane components of polarization. Also, the compressive strain and/or interfacial charge discontinuity might

increase the out-of-plane component of polarization.

Although finding all the domains is challenging and time-consuming, we found and observed all kinds of four domains with STEM (Fig. R14b below). Domain #1 shows an obvious out-of-plane polar axis, and this result well agrees with SHG and theoretical prediction. Additionally, other domains #2 and #3 show randomly arranged weak polarization, since the polarization has a negligible projection along our zone axis in these domains. We added these data in the revised Supplementary Information as Supplementary Fig. S16.

Fig. R14 | ABF-STEM observation of domain structures of $R3c$ - CaTiO_3 on LaAlO_3 (111)_{pc} substrate. a, Schematic of four possible domains of $R3c$ - CaTiO_3 . Black arrows indicate the Ti polar displacement. **b**, Atomic scale imaging and polarization mapping on four domains of $R3c$ - CaTiO_3 via ABF-STEM.

Question#3

Also, the equivalent polar axes would lead to domain formation. In Fig. 4, the authors showed the vertical PFM images of domain structures written with applied voltage. It'd be informative to show what the pristine state of the sample is without external field applied. Is there out-of-plane domain observed?

I also suggest the authors perform lateral PFM to map out the in-plane polarization distribution.

Response#3

First of all, it should be noted that the primary interest of our PFM study is 2.6 nm-thick CaTiO_3

films. For such thin films, conventional PFM measurement cannot resolve the ferroelectric domains for large signal-to-noise ratio. To overcome this, we did dual AC resonance tracking (DART) mode, where the PFM is performed near the electro-mechanical resonance of samples [B. J. Rodriguez, C. Callahan, S. V. Kalinin, and R. Proksch, *Nanotechnology* **18** 475504 (2007)]. But, in our set-up, this technique is only available in the vertical PFM, not in the lateral PFM. The pristine domain of 2.6 nm-thick *R3c*-CaTiO₃ is polarized downward and we cannot resolve the in-plane domain in the lateral PFM (Fig. R15).

Fig. R15 | In-plane PFM imaging of *R3c*-CaTiO₃. **a**, Bipolar, out-of-plane domain pattern of *R3c*-CaTiO₃ imaged by Dual AC resonance tracking (DART) PFM. **b**, Bipolar, in-plane domain pattern of *R3c*-CaTiO₃ imaged by vector PFM mode.

Question#4

3. It is known that epitaxial strain relaxes with film thickness. Have the authors examined thicker films to explore the critical thickness where the enforced polar distortion disappears?

Response#4

We thank the reviewer for the interesting comment on our work. Following the reviewer's comment, we examined thicker CaTiO₃ films using the ABF-STEM technique (Figs. R6 and R7 in our responses to Reviewer #1's comments). As the reviewer expected, CaTiO₃ becomes nonpolar *Pnma* structure at the critical thickness around 8 nm. Unexpectedly, the *Pnma* CaTiO₃ film was still epitaxially strained on the LaAlO₃ (111) substrate (Fig. R4 in our responses to Reviewer #1's comments). This suggests that atomic-structure relaxation of OOR occurs separately from the epitaxial strain relaxation in CaTiO₃/LaAlO₃ (111) heterostructures, consistent with an earlier report on the other systems [J. Fowlie, *et al.*, *Nano. Lett.* **19**, 4188-4194 (2019)].

Question#5

4. Page 7, second paragraph: "Bipolar domain patterns (Fig. 4c,d), created in CaTiO₃ (111) films, also remain stable for at least several hours." Please avoid such vague description and give the details of time-dependence PFM experiments, i.e., the time duration, time-dependence

of the signal strength, etc. Also, the authors should note that there are relaxor systems where the PFM response for the written domains can last for days, e.g., Moghadam et al., Nano Lett., 17, 6248 (2017).

Response#5

We really appreciate the reviewer's valuable comment on our mistake. We carried out a systematic time-dependent PFM study (Fig. R16 below). We observed the PFM signals decays in the power law with an exponent of 0.145. Although it is decaying, the behavior is comparable with the other prototypical ferroelectrics in the thin film forms [D. Lee *et al.*, *Science* **349**, 1314-1317 (2015)].

We thank the reviewer for the interesting comment on the relaxor system. According to the paper mentioned by the reviewer, 15 nm-thick relaxor system shows a decay exponent of 0.22. It is surprising that our much thinner (i.e., 2.6 nm-thick) CaTiO₃ film shows a much smaller decay exponent of 0.145 (i.e., more stable polarization). Therefore, our PFM result confirms the robust room-temperature ferroelectricity in CaTiO₃ (111) films, consistent with our DFT calculation, SHG data, and STEM imaging results.

Fig. R16 | Stability of ferroelectric domain of R3c-CaTiO₃. **a**, Bipolar domain of R3c-CaTiO₃ which persists for an hour. **b**, Time-dependent PFM amplitude of R3c-CaTiO₃ which follows a power-law with an exponent of 0.145.

REVIEWER COMMENTS

Reviewer #1 (Remarks to the Author):

The authors have answered all my questions and those of the other referees conscientiously and thoroughly. I am not yet fully satisfied with the answer on the oxygen vacancy content. Rather than comparing their PLD films with MBE grown films, which could also grow with oxygen off-stoichiometry, it would have been more convincing to compare their lattice parameters with those of the bulk material or to show the effect of oxygen pressure or time during the post-annealing or cooling down process on the lattice parameters. Nevertheless, with the new information presented in the revised version, it is now much more convincing that strain alone is not responsible for the stabilization of the R3c phase. In addition, nominal compressive strain is not likely to induce large numbers of oxygen vacancies. Thus, I find this new version convincing. This is a solid, well-written and original paper and I can advise its publication in Nature Communications.

Reviewer #2 (Remarks to the Author):

We thank the authors for their detailed reply. The paper is now clearer and can be considered for publication.

Reviewer #3 (Remarks to the Author):

In the revised manuscript, the authors have addressed most of my previous comments. However, there are a couple of points that require further clarification, as detailed below:

1. The interpretation of SHG results shown in Fig. 2d seems to be problematic. The polar plot of the SHG signal shows a clear 3-fold symmetry. I'm very confused by the authors' claim that this is consistent with an out-of-plane polar axis ([111], Domain #1 in Fig. R8). I assume the incident light is along the film surface normal, which means it's only sensitive to the in-plane symmetry breaking. In this case, the out-of-plane polar axis would not generate any SHG response. In fact, the three-fold in-plane polar symmetry revealed by the SHG signal can be naturally accounted for by the [110] polar axis (Domain #4), which will lead to domain formation in the (111)-oriented film. Can the authors elaborate on this point? Also, please specify the SHG measurement geometry, e.g., incident angle, optical setup for the perpendicular/parallel SHG, in the Method section.
2. The PFM results shown in Fig. R15 reveal clearly that the pristine film is not in a uniformly polarized state. It may also provide evidence to the coexistence of different domain structures in this system. I think this data should be included in the supplementary and be properly discussed in the main text.
3. The time-dependence of the PFM signal does not rule out the possibility that the CTO sample is a relaxor. The extracted decay exponent is also comparable with those of PMN-PT (PRL 104, 197601 (2010)) and strained SrTiO₃ (Ref. 28), which are known to be relaxors. I suggest the authors leave this possibility open in the discussion.

Responses to the comments of Reviewer #1

The authors have answered all my questions and those of the other referees conscientiously and thoroughly. ... Thus, I find this new version convincing. This is a solid, well-written and original paper and I can advise its publication in Nature Communications.

We thank the reviewer for recommending publication of our work in *Nature Communications*, as well as for the remark that our paper is solid, well-written, and original.

Question#1

I am not yet fully satisfied with the answer on the oxygen vacancy content. Rather than comparing their PLD films with MBE grown films, which could also grow with oxygen off-stoichiometry, it would have been more convincing to compare their lattice parameters with those of the bulk material or to show the effect of oxygen pressure or time during the post-annealing or cooling down process on the lattice parameters.

Response#1

We thank the reviewer for the valuable comment. Following the reviewer's suggestions, we compared the lattice parameters of bulk CaTiO₃, a pristine CaTiO₃ film grown on an LSAT (001) substrate, and a post-annealed (in oxygen) CaTiO₃/LSAT (001) film (Fig. R1 below). First, the pristine CaTiO₃/LSAT (001) film has a smaller out-of-plane *c* lattice parameter of 0.3798 nm compared to bulk CaTiO₃ (pseudocubic lattice $a_{pc} = 0.3822$ nm). Considering that CaTiO₃ experiences a tensile strain of +1.2% on the LSAT (001) substrate with $a = 0.3868$ nm, the contraction of the *c* lattice parameter is reasonable. Assuming Poisson's ratio of $\nu = 0.23$ (as in SrTiO₃), the biaxial in-plane strain $\epsilon_{xx} = +1.2\%$ would cause an out-of-plane strain of $\epsilon_{zz} = -[2\nu/(1 - \nu)]\epsilon_{xx} = -0.72\%$, yielding a *c* lattice constant of 0.3795 nm. This is in good agreement with the experimentally measured value (i.e., 0.3798 nm). Second, we examined the effect of post-annealing in oxygen on the lattice parameter of the CaTiO₃/LSAT (001) film. The post-annealing temperature, background gas pressure, and annealing time were 500°C, 760 Torr of pure oxygen, and 2 h, respectively. After the post-annealing process, the lattice parameter of the CaTiO₃/LSAT (001) film remained unchanged. This indicated that our optimal CaTiO₃ was already well-oxidized during growth, with post-annealing having little effect on the CaTiO₃ film. We added this information in the Methods section of the revised manuscript, with the data presented as Supplementary Fig. S7g.

Fig. R1 | Lattice parameters of an optimal CaTiO_3 thin film grown on an LSAT (001) substrate. X-ray θ - 2θ scan of a 20 nm-thick CaTiO_3 thin film grown on a $(\text{LaAlO}_3)_{0.3}$ - $(\text{Sr}_2\text{AlTaO}_6)_{0.7}$ (LSAT) (001) substrate with an optimal laser energy density. The black and red curves correspond to the pristine CaTiO_3 film and the post-annealed (in an oxygen environment) CaTiO_3 film, respectively.

Question#2

Nevertheless, with the new information presented in the revised version, it is now much more convincing that strain alone is not responsible for the stabilization of the R3c phase. In addition, nominal compressive strain is not likely to induce large numbers of oxygen vacancies.

Response#2

We appreciate the reviewer's positive evaluation of our revised manuscript. As correctly pointed out by the reviewer, it is well accepted that compressive strain tends to increase the formation energy of oxygen vacancies in oxide perovskites; therefore, the compressively strained CaTiO_3 on a LaAlO_3 substrate would be free of excessive oxygen vacancies.

Responses to the comments of Reviewer #2

We thank the authors for their detailed reply. The paper is now clearer and can be considered for publication.

We thank the reviewer for recommending publication of our work in *Nature Communications*, as well as for the remark that our paper is now clearer.

Responses to the comments of Reviewer #3

In the revised manuscript, the authors have addressed most of my previous comments. However, there are a couple of points that require further clarification, as detailed below:

We thank the reviewer for stating that we had addressed most of his/her previous comments. Below, we address the reviewer's remaining concerns.

Question#1

1. The interpretation of SHG results shown in Fig. 2d seems to be problematic. The polar plot of the SHG signal shows a clear 3-fold symmetry. I'm very confused by the authors' claim that this is consistent with an out-of-plane polar axis ([111], Domain #1 in Fig. R8). I assume the incident light is along the film surface normal, which means it's only sensitive to the in-plane symmetry breaking. In this case, the out-of-plane polar axis would not generate any SHG response. In fact, the three-fold in-plane polar symmetry revealed by the SHG signal can be naturally accounted for by the [110] polar axis (Domain #4), which will lead to domain formation in the (111)-oriented film. Can the authors elaborate on this point? Also, please specify the SHG measurement geometry, e.g., incident angle, optical setup for the perpendicular/parallel SHG, in the Method section.

Response#1

We are grateful to the reviewer for pointing out insufficient explanation of the SHG experiment. All SHG was carried out with an incidence angle of 45°, as opposed to normal incidence. We apologize for this misunderstanding. Our experimental set-up is illustrated in Fig. R2. “p” (“s”) indicates parallel (perpendicular) polarization with respect to the plane of incidence. As a result, “p” reflects both out-of-plane and in-plane inversion symmetry breaking, whereas “s” reflects only in-plane inversion symmetry breaking.

For the interpretation of our data, we assume Domain #1 in Fig. S18 with out-of-plane polarization. The allowed nonlinear susceptibility (χ) components of the $3m$ point group are $\chi_{xxx} = -\chi_{xyy} = -\chi_{yyx} = -\chi_{yxy}$, $\chi_{zxx} = \chi_{zyy}$, $\chi_{xxz} = \chi_{yyz}$, $\chi_{zxz} = \chi_{zyz}$, and χ_{zzz} . Using the formula below, we calculated and fitted the polarization-dependent SHG intensity from CaTiO₃ films.

$$I(2\omega) = \left| \chi_{ijk} E_j^\omega E_k^\omega \right|^2,$$
$$I_{pp}(2\omega) = \left[(\chi_{zyy} - \chi_{yzy} - \chi_{yyz} + \chi_{zzz}) + \chi_{xxx} \sin(3\varphi) \right]^2,$$
$$I_{sp}(2\omega) = 4 \left[\chi_{zyy} - \chi_{xxx} \sin(3\varphi) \right]^2,$$
$$I_{ps}(2\omega) = \left[\chi_{xxx} \cos(3\varphi) \right]^2, \quad I_{ss}(2\omega) = 4 \left[\chi_{xxx} \cos(3\varphi) \right]^2$$

We understand the reviewer’s confusion, as the out-of-plane polarization alone cannot explain the observed SHG signals in ps and ss geometries, which reflect in-plane inversion symmetry breaking (Fig. 2d in the main text). Here, what breaks the in-plane inversion symmetry is not the electrical polarization, but the absence of the crystallographic mirror symmetry of the $R3c$ structure along the in-plane direction. We added this important discussion in the Methods section of the revised manuscript, as well as an illustration of the experimental set-up as Supplementary Fig. S10.

Fig. R2 | Experimental set-up for reflection second harmonic generation. Reflection second harmonic generation in which the CaTiO_3 thin film is irradiated with an 800-nm femtosecond wave at an incidence angle of 45° . “p” (“s”) indicates parallel (perpendicular) polarization with respect to the plane of incidence.

Question#2

2. The PFM results shown in Fig. R15 reveal clearly that the pristine film is not in a uniformly polarized state. It may also provide evidence to the coexistence of different domain structures in this system. I think this data should be included in the supplementary and be properly discussed in the main text.

Response#2

We agree with the reviewer’s interpretation of our PFM data. The coexistence of different domain structures would result in complex PFM signals from the pristine CaTiO_3 films, and it is consistent with our STEM observations of the four domains (Fig. S18). Following the reviewer’s comment, we added the previous Fig. R15 as Supplementary Fig. S14 and included a more detailed discussion of these findings in the revised manuscript (Line #164), as given below.

The pristine domain structure of CaTiO_3 (111) films was difficult to define (Supplementary Fig. S14); however, it may be attributable to the ferroelectric domain structure (Supplementary Fig. S18), which can assume a mono-domain configuration upon electrical poling (Fig. 4c,d).

Question#3

3. The time-dependence of the PFM signal does not rule out the possibility that the CTO sample is a relaxor. The extracted decay exponent is also comparable with those of PMN-PT (PRL 104, 197601 (2010)) and strained SrTiO₃ (Ref. 28), which are known to be relaxors. I suggest the authors leave this possibility open in the discussion.

Response#3

We thank the reviewer for the suggestion. We would like to emphasize that according to our DFT calculations, the ferroelectricity of *R3c* CaTiO₃ could be distinct from those of relaxors. While relaxor ferroelectrics show critical reliance on local disorders (e.g., PMN-PT: inhomogeneities of chemical composition and local symmetry, SrTiO₃ [ref. 28]: Sr deficiency-induced polar nanoregions), “pure” *R3c* CaTiO₃ is theoretically predicted to show robust ferroelectricity, as confirmed in STEM experiments. Therefore, we believe that the polarization relaxation observed in our 2.6 nm-thick CaTiO₃ is just a typical characteristic of ultrathin ferroelectrics, stemming from the depolarization field. However, as commented on by the reviewer, an additional discussion of other possible explanations regarding polarization relaxation is necessary. Accordingly, we added a discussion of relaxor ferroelectrics in the revised manuscript (Line #170), as given below.

*Although comparable decay exponents have been reported in the relaxor ferroelectrics²⁸, a similar relaxation has also been observed in prototypical ferroelectric films at the ultrathin limit²⁹. Based on our comprehensive DFT and STEM studies, we attribute the relaxation behavior of the ferroelectric *R3c*-CaTiO₃ to the huge depolarization field in ultrathin ferroelectric films.*

REVIEWERS' COMMENTS

Reviewer #3 (Remarks to the Author):

In the revised manuscript, the authors have thoroughly addressed all my comments/questions. I can recommend its publication in Nature Communications.

Responses to the comments of Reviewer #3

In the revised manuscript, the authors have thoroughly addressed all my comments/questions. I can recommend its publication in Nature Communications.

We thank the reviewer for recommending publication of our work in *Nature Communications*, as well as for the remark that we have thoroughly addressed all his/her comments/questions.